# DL-BENCH: DEEP LEARNING SPECIFIC CODE GENERATION BENCHMARK

## ABSTRACT

Deep learning (DL) has revolutionized areas such as computer vision, natural language processing, and more. However, developing DL systems is challenging due to the complexity of DL workflows. Large Language Models (LLMs), such as GPT, Deepseek, Claude, Llama, Mistral, Qwen, etc., have emerged as promising tools to assist in DL code generation, offering potential solutions to these challenges. Despite this, existing benchmarks like DS-1000 are limited, as they primarily focus on small DL code snippets related to pre/post-processing tasks and lack comprehensive coverage of the full DL pipeline, including different DL phases and input data types. Similarly, MLE-bench focuses more on Machine Learning Engineering (MLE) tasks and broader ML workflows, without leveraging test cases.

To address this, we introduce DL-Bench, a novel benchmark dataset designed for function-level DL code generation. DL-Bench categorizes DL problems based on three key aspects: phases such as pre-processing, model construction, and training; tasks, including classification, regression, and recommendation; and input data types such as tabular, image, and text. DL-Bench diverges from related benchmarks, DS-1000 and AICoderEval, across four dimensions: it occupies a semantically distinct region for both prompts and code embedding, emphasizes DL constructs with a higher DL/ML token ratio, and requires more complex code solutions. State-of-the-art LLMs (e.g., O3-Mini, DeepSeak-V3) achieve, on average, significantly lower 28.5% pass@1 score on DL-Bench than on DS-1000 (53.3%). This result underscores DL-Bench's greater challenging problems set. Our taxonomy of bugs found in LLM-generated DL code highlights the distinct challenges that LLMs face when generating DL code compared to general code. Furthermore, our analysis reveals substantial performance variations across categories which emphasizes valuable insights that DL-Bench offers for potential improvement in the DL-specific generation. Our preliminary result shows that DL-Bench can enhance LLM performance as a categorization training dataset, achieving an average 4.2% improvement on DS-1000 with guided three-shot learning.

Overall, our empirical results demonstrate the utility of DL-Bench as a comprehensive benchmark while offering insights for future improvements across diverse functional categories.

## 1 INTRODUCTION

In recent years, machine learning (ML) and deep learning (DL) have advanced significantly and have been integrated into various fields Hordri et al. (2016); Kamilaris & Prenafeta-Boldú (2018); Gamboa (2017). DL coding has its challenges Arpteg et al. (2018), and because of its widespread use, many DL systems are developed by domain experts who are often not software developers Park et al. (2021); Singaravel et al. (2020), which amplifies the problems even more.

Recently, with the rise of Large Language Models (LLMs) such as ChatGPT, LLMs are considered among the best solutions for coding tasks Wang et al. (2021); Feng et al. (2020); Achiam et al. (2023) as demonstrated by numerous code generation benchmark datasets.However, until recently, most of these benchmarks focused on general programming tasks. Shin et al. (2023) are the first to underline the distinct challenges of generating ML/DL code compared to general code. However, their generated code evaluation relies on less suitable similarity metrics as very different code snippets can have the same functionality, and a small change in a code snippet can drastically alter its semantics.

A few datasets, such as MLE-bench Chan et al. (2024) or AICoderEval Xia et al. (2024), offer examples of ML-specific code generation on the ML workflow level, which does not fit the LLM's usage, where developers need help with generating specific functions. These benchmarks often evaluate LLMs based on the final ML system's performance (e.g., accuracy, F1, etc.). Among these, DS-1000 Lai et al. (2023) provides small (a few lines) ML-specific code snippets, primarily focused on pre/post-processing tasks. It also does not provide any categorizations, such as ML tasks, DL phases, or input types, which could provide valuable insights for code generation improvement.

To address these gaps, we introduce DL-Bench, a novel dataset designed to benchmark DL-specific code generation at a functional level. Each entry includes the code generation prompt, the ground-truth code at the function level, and an extensive set of unit tests. Unlike DS-1000 and MLE-bench, DL-Bench provides a more comprehensive and diverse set of function-level samples that cover all phases in the DL pipeline for various ML tasks and input data types. These entries are categorized into three aspects: (1)*The DL/ML pipeline stages*: pre/post-processing, model construction, training, inference, and evaluation, (2)*The DL/ML tasks*: classification, object detection, image segmentation, time-series prediction, recommendation, and regression, and (3)*The input data types*: text, image, and array. These categorizations enable a more in-depth evaluation and analysis of future techniques in generating DL-specific code.

We qualitatively compare DL-Bench with its most related benchmarks (DS-1000 and AICoderEval) by examining four aspects of dataset divergence. First, we show that DL-Bench occupies a semantically distinct region of the embedding space, hence contains novel problem domains and different solution patterns. Second, we reveal that DL-Bench emphasizes DL constructs heavily. Third, we demonstrate that DL-Bench problems require more complex solutions, hence are more challenging for LLMs. Finally, state-of-the-art LLMs (e.g., O3-Mini, DeepSeak-V3) struggle to solve DL-Bench's problems with significantly lower **28.5%** pass@1 score on DL-Bench than on DS-1000 (53.3%).

Furthermore, our qualitative analysis indicates that the difficulty of generating code varies significantly across categories. For example, O3-Mini reaches an accuracy of 39.4% for pre/post-processing tasks but only 30.4% for model construction. The pass@1 rate varies even more among task types, ranging from 53.1% for recommendation tasks to 26.3% for segmentation tasks on O3-Mini. These large gaps in performance across categories highlight the importance of insights that DL-Bench can bring to help improve the LLM DL code generation capability. Additionally, we construct a bug taxonomy of the issues found in the generated DL code. When compared to LLM-generated general code, LLM-generated DL code exhibits a higher frequency of *deviation from the prompt* issues and a new issues category *arithmetic and logical errors*.

Finally, we demonstrate a potential usage where DL-Bench can be used to guide few-shot prompting. In this usage, DL-Bench, on average, can consistently improve represented LLMs by **4.2%** on DS-1000. DL-Bench's data is available in our Kaggle repository[1]. The evaluation code is also available in our GitHub repository[2].

## 2 RELATED WORKS

There are multiple benchmarks that contain code samples for data science tasks, such as JuICe Agashe et al. (2019), PandasEval and NumPyEval Zan et al. (2022), and JuPyT5 Chandel et al. (2022). None of them contains any test cases, so similarity metrics such as the BLEU score are used for evaluation of generated code. Unlike these benchmarks, DL-Bench contains multiple test cases for each entry, which enable better evaluation metrics such as pass@1 for generated code. Shin et al.Shin et al. (2023) explore the effectiveness of neural code generation by selecting ML/DL-specific samples from JuICeAgashe et al. (2019). However, similar to JuICe, they evaluate generated code using similarity metrics, which is not suitable for generated code evaluation. DL-Bench contains test cases that better evaluate the correctness of the generated code. Recently, MLE-bench Chan et al. (2024) contains ML engineering workflows in Kaggle competitions. Similarly, AICoderEval Xia et al. (2024) presents a broader ML workflow benchmark. These workflow-level benchmarks focus on complete solutions and do not provide evaluations of approaches that serve developers who need a specific function.

---

[1]https://kaggle.com/datasets/b4b26b3d3ffe9930789d43da1377265a445add5023f87c9dc4bfcf4b50f93a62
[2]https://anonymous.4open.science/r/DL-Bench-71ED/

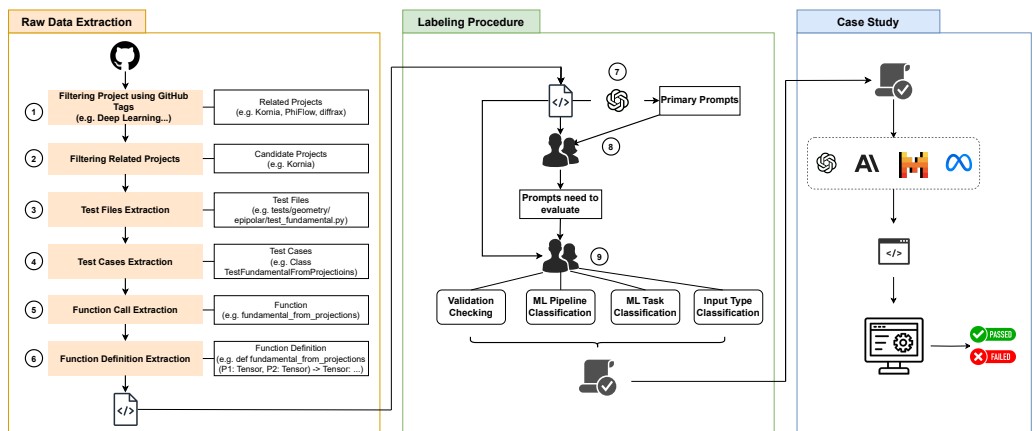

Figure 1: DL-Bench construction procedure

> Create a prompt using the provided code and its docstring, incorporating the function or class name, inputs, and outputs.
>
> 1. Code: [CODE]  2. Docstring: [DOCSTRING]

Figure 2: Template of generating prompt from code

DS-1000 Lai et al. (2023) contains problems sourced from StackOverflow for localized data science tasks such as pre-/post-processing.

There have been multiple general code generation benchmarks such as HumanEval Chen et al. (2021), AiXBench Hao et al. (2022),MultiPL-E Cassano et al. (2022),MBPP Austin et al. (2021),Spider benchmark Yu et al. (2018),CoderEval Yu et al. (2024),APPS benchmark Hendrycks et al. (2021), and RepoEval Zhang et al. (2023). All the above-mentioned benchmarks focus on general programming.

DL-Bench differs from prior work in three key aspects: (1) it focuses on ML/DL tasks rather than general data science or ML engineering, (2) we categorize the data by ML phases, task types, and data types, and (3) our granularity is at the function level rather than at the script or workflow level. For example, one of our prompts instructs the generation of a `maximum_weight_matching` function, which performs a precise weight matching operation tailored to a DL-specific need. Moreover, unlike the other datasets, DL-Bench is based on GitHub repositories containing real code and tests.

## 3 BENCHMARK CONSTRUCTION

DL-Bench consists of 520 instances of AI and DL data points (filtered from over 2,000 raw data points). The data is curated from 30 GitHub repositories (selected from an initial pool of 160 related repositories). DL-Bench is released with a GNU license to ensure legal usage of code from these 30 repositories.

The construction process of DL-Bench consists of two main phases: The Raw Data Extraction and the Labeling Procedure. The raw data extraction involves six semi-automatic steps. Since DL-Bench is designed to have diverse and realistic code samples, the first step ① is to construct DL-Bench from code crawled from highly rated GitHub repositories (i.e., with the most stars), updated after the training cutoff of GPT-4o to mitigate data leakage, filtered using 30 DL-related terms such as "neural-networks", "pytorch", "computer-vision". We then manually select (step ②) 160 high quality candidate DL projects (i.e., involve the integration of DL and AI-related frameworks, comprehensive test cases, clear and well-written docstrings, and detailed contribution guidelines). We then employed a bespoke utility to extract the test files and then test cases from each repository (step ③ and ④). By performing static analysis, we were able to track and collect all of the functions under test in step ⑤ to form the raw data that is the base of DL-Bench.

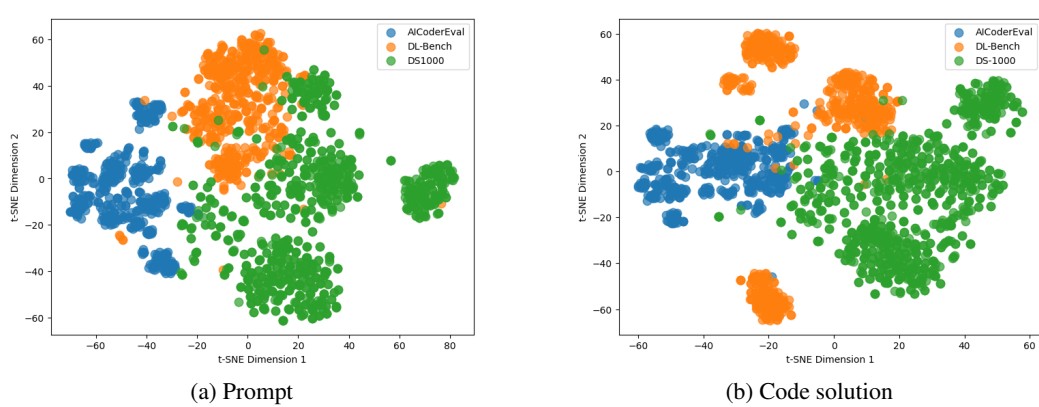

(a) Prompt                                    (b) Code solution

Figure 3: Prompt and code solution embeddings projection for DL-Bench, DS-1000, and AICoderEval

Once the raw data is extracted, the labeling procedure starts. To speed up the task of constructing the prompt for each code sample, we utilize LLM (i.e., GPT-4o) as a code-explanation toolNam et al. (2024) to generate the first prompt candidate for each function under test (step ⑦). Four co-authors were then tasked with manually filtering (step ⑧) each entry to ensure that each function is highly relevant (i.e., contributes to a DL task such as image recognition, utilizes at least one recognized DL framework, and implements a relatively advanced and sophisticated algorithm). Finally, we conduct a manual labeling process involving four co-authors (step ⑨) to refine the prompt and label each code sample with the appropriate category from our three chosen types of categories: DL pipeline phases, ML task types, and input types. Due to space limitations, a more detailed description of each step is included in the appendix.

## 4   QUANTITATIVE ANALYSIS

To differentiate DL-Bench from prior benchmarks and demonstrate its potential, we perform a quantitative comparison between DL-Bench and its related benchmarks (DS-1000 and AICoderEval). We first analyze the data in each benchmark to show that DL-Bench contains novel and challenging DL-specific problems that require more complex solutions. Then we empirically show that DL-Bench is more challenging to solve than DS-1000 by comparing the performance of representative LLMs.

### 4.1   DL-BENCH CONTAINS DISTINCTIVE AND MORE CHALLENGING PROBLEMS AND SOLUTIONS WHEN COMPARED TO DS-1000 AND AICODEREVAL.

In this section, we evaluate how DL-Bench diverges from its closest data-science and ML benchmarks, DS-1000 and AICoderEval. First, we contrast the semantic spaces of their input prompts and code solutions by comparing embedding distributions. Second, we gauge each benchmark's DL orientation by tracking the prevalence of DL-specific tokens in the reference code Finally, we measure code complexity to provide a holistic view of DL-Bench's problems relative difficulty.

**Semantic Comparison:** To compare the semantic prompt space, we embed each natural-language prompt with the `all-MiniLM-L6-v2` sentence-transformer Li et al. (2020). The average cosine similarity values between DL-Bench and related benchmarks are relatively low (**0.188** for DS-1000 and **0.184** for AICoderEval). Such notable semantic divergence in DL-Bench's input prompts from related benchmarks indicates that DL-Bench covers distinct domains or task formulations. Additionally, when projecting the embeddings into two dimensions using t-SNE Van der Maaten & Hinton (2008) (Figure 3a), the visualization reveals separable clusters corresponding to DL-Bench and DS-1000. This distinct clustering further supports the semantic uniqueness of DL-Bench and highlights its complementary role in benchmark diversity.

To demonstrate the distinctiveness of DL-Bench's solutions, we compare the semantic representation of its ground-truth code with that of DS-1000 and AICoderEval. To this end, we use `CodeBERT` Feng et al. (2020) to generate code embeddings for each reference implementation, and apply cosine similarity computation. DL-Bench problems require code solutions that are semantically different from DS-1000 and AICoderEval, with the average cosine similarity of **0.538** and **0.638** respectively. This indicates each benchmark covers a distinct set of tasks and requires different solution patterns. Figure 3b visualizes these differences by applying t-SNE to the ground-truth code embedding space.

Table 1: Pass@1 (%) scores for various SOTA LLMs on DS-1000 and DL-Bench.

| Benchmark | O3-Mini | DeepSeek-V3 | GPT-4o | Claude 3.5 Sonnet | Llama 3.1 70B | Mixtral 8*22B | QwenCoder | Avg. |
|---|---|---|---|---|---|---|---|---|
| DL-Bench | **35.1** | 30.5 | 30.2 | 30.5 | 26.7 | 23.9 | 22.8 | 28.5 |
| DS-1000 | 61.0 | 61.7 | 51.1 | **61.9** | 40.9 | 39.3 | 57.3 | 53.3 |

The projection reveals separated clusters for DL-Bench, DS-1000, and AICoderEval, providing further evidence that DL-Bench offers complementary coverage and contributes novel content to existing DL/ML code generation benchmarks.

**DL-relevance Analysis:** To measure DL-Bench's DL/ML relevancy, we compute a domain-relevance metric based on the DL/ML tokens ratio in the reference code. Averaged across instances, DL-Bench attains a ratio of **0.785**, far exceeding DS-1000 (0.131) and AICoderEval (0.437). Put differently, more than three-quarters of the lexical footprint in DL-Bench code is devoted to DL/ML concepts, whereas only one-eighth in DS-1000 and less than half in AICoderEval references such terms.

**Solution Complexity:** To gauge the relative complexity of DL-Bench's problems, we compare three structural metrics: lines of code (LOC), cyclomatic complexity, and cognitive complexity (extracted with *radon* Lacchia (2025)). On average, solutions in DL-Bench span **14.8 LOC**, nearly double AIcoderEval (8.5) and more than quadruple DS-1000 (3.6). Cognitive complexity follows a similar pattern (**4.26** vs. 0.31 and 0.008), underscoring more complex nested structures and longer call chains in DL-Bench.

> **Finding 1:** DL-Bench's problems focus on DL-specific domain and occupy a distinct semantic space. Furthermore, DL-Bench contains difficult problems that require significantly more complex code solutions that pose significant challenges to advanced LLMs.

### 4.2 Performances of SOTA LLMs on DL-Bench and DS-1000

This analysis investigates how the existing ML code generation benchmark (DS-1000) and DL-Bench evaluate seven representative LLMs covering a spectrum of parameter scales, licensing regimes, and training specializations. Since AICoderEval has not been published and does not provide sufficient and reliable evaluation scripts, we have decided to exclude it from this evaluation. A commonly used pass@k Lyu et al. (2024), which measures the likelihood that at least one of the k-generated solutions passes all test cases, is used in this evaluation. To minimize non-determinism and improve reproducibility, we set the temperature to zero for all LLMs Bommasani et al. (2021). We also run the experiment on DL-Bench five times, and the standard deviation is small between 0.7% and 1.8%, indicating that the zero-temperature induces more stable performance for comparison. We intentionally avoided using specialized prompt strategies, opting instead for vanilla prompts to focus on the model's baseline performance. However, the use of advanced prompt engineering strategies could yield different results. In a later section, we demonstrate a potential usage of DL-Bench as a guided few-shot dataset. Table 1 shows the pass@1 of SOTA LLMs on DL-Bench and DS-1000.

Our evaluation shows that even the most advanced model, such as O3-Mini, struggles with ML/DL-specific code generation. Specifically, O3-Mini achieves 61.0% pass@1 in DS-1000 but only 35.1% pass@1 on DL-Bench. Similarly, all other tested LLMs get much lower pass@1 scores in DL-Bench than DS-1000. We also compute pass@3 and pass@5 of the seven LLMs on DL-Bench (complete table is provided in the Appendix). O3-mini benefits the most when having additional candidates; however, its performance on DL-bench is still low at 40.2% pass@5 rate. The overall weak performance of these models highlights the ongoing challenges in generating reliable, executable ML/DL-specific code, supporting the need for deeper analysis to identify problematic areas that DL-Bench can provide.

Our separability and ranking agreement analysis between DL-Bench and DS-1000 yielded a huge (more than 2.0) Cohen's-d effect size of **3.13** and a large (more than 1.0) Fisher's ratio of **4.9**, confirming that DL-Bench is markedly more challenging with significantly lower and distinct distribution of pass@1 scores. However, the average Spearman correlation of **0.50** (p = 0.25) shows that the ranking of models is moderately consistent with DS-1000. This suggests that DL-Bench presents harder problems and contains additional aspects that can capture slightly different relative performance among LLMs.

To analyze the effect of data leakage, we experiment with "live versions" of DL-Bench. Table 2 shows the SOTA LLMs' performance on DL-Bench with different cutoff dates. As the recency of data increases, the models' performance declines. This result indicates that more recent data is less

Table 2: Pass@1 (%) scores for various SOTA LLMs on Live DL-Bench

| Model | Overall | After Oct 2023 | After Jan 2024 | After May 2024 | After Sep 2024 |
|---|---|---|---|---|---|
| Claude 3.5 Sonnet | 30.5 | 30.4 | 30.0 | 28.3 | 27.6 |
| DeepSeek V3 | 30.5 | 31.4 | 29.6 | 27.3 | 27.5 |
| GPT-4o | 30.2 | 31.4 | 29.5 | 26.3 | 25.7 |
| LLaMA 3.1 70B | 26.7 | 27.8 | 27.5 | 26.1 | 25.0 |
| Mistral 8×22B | 23.9 | 24.4 | 23.8 | 22.6 | 23.1 |
| O3-mini | 35.1 | 35.8 | 32.8 | 29.6 | 30.5 |
| Qwen Coder 2.5 | 22.8 | 23.6 | 22.7 | 23.6 | 24.2 |

Table 3: Pass@1 (%) scores on DL-Bench across stages, ML/DL tasks, and input data types

| Category | O3-Mini | DeepSeek-V3 | GPT-4o | Claude 3.5 Sonnet | Llama 3.1 70B | Mixtral 8×22B | QwenCoder | Avg. |
|---|---|---|---|---|---|---|---|---|
| Stages in pipeline | | | | | | | | |
| Pre/Post Processing | **39.4** | 33.9 | 34.5 | 33.2 | 30.2 | 27.3 | 24.6 | **31.9** |
| Model Construction | **30.4** | 26.7 | 23.9 | 24.4 | 19.9 | 16.8 | 14.2 | 22.3 |
| Training | **31.2** | 28.4 | 30.4 | 26.2 | 28.0 | 26.6 | 27.4 | 28.3 |
| Inference | **38.4** | 26.4 | 28.9 | 27.1 | 26.1 | 26.9 | 23.4 | 28.1 |
| Evaluation & Metrics | **35.6** | 28.6 | 25.4 | 31.2 | 24.7 | 23.9 | 23.8 | 27.6 |
| ML tasks | | | | | | | | |
| Classification | **35.9** | 25.7 | 27.6 | 28.6 | 23.5 | 29.0 | 23.1 | 27.6 |
| Regression | **40.0** | 20.8 | 26.5 | 26.9 | 11.8 | 20.9 | 12.8 | 22.8 |
| Object Detection | **29.8** | 21.2 | 27.7 | 20.2 | 10.8 | 9.7 | 9.4 | 18.4 |
| Image Segmentation | 26.3 | **27.1** | 13.8 | 17.0 | 19.2 | 14.2 | 21.4 | 19.8 |
| Time Series Prediction | **38.8** | 19.3 | 35.4 | 27.5 | 19.3 | 19.3 | 19.3 | 25.5 |
| Recommendation | 53.1 | 34.4 | 45.2 | **56.9** | 33.4 | 45.7 | 39.4 | **44.1** |
| General | **35.7** | 33.0 | 31.4 | 29.9 | 31.0 | 26.8 | 22.8 | 30.1 |
| Input data types | | | | | | | | |
| Image | **33.5** | 30.1 | 27.6 | 25.9 | 21.8 | 18.7 | 16.8 | 24.9 |
| Text | **51.8** | 27.6 | 39.1 | 43.7 | 33.7 | 43.7 | 27.6 | **38.1** |
| Structured Array | **36.9** | 28.3 | 27.3 | 28.5 | 24.9 | 28.5 | 21.2 | 27.9 |
| Others | **34.5** | 30.9 | 33.6 | 30.9 | 32.0 | 28.6 | 28.9 | 31.3 |

likely to be leaked and pose greater challenges for LLMs. To mitigate the effect of data leakage, we plan to add more "live versions" of DL-Bench in the future.

> **Finding 2:** Our evaluation indicates that current SOTA LLMs struggle to generate correct, executable code for ML/DL tasks with an average pass@1 score of **28.5%** on DL-Bench. Although O3-Mini is the strongest among the tested models, it still falls short of meeting practical standards with a pass@1 score of only **35.1%**. Empirically, DL-Bench presents more challenging problems and contains different aspects that captures a slightly different ranking among LLMs.

## 5 QUALITATIVE ANALYSIS

This section provides a deeper analysis of which kinds of DL-specific code are harder to generate, and the common issues that generated DL-specific code has.

### 5.1 WHICH KINDS OF DL-SPECIFIC CODE POSE A GREATER CHALLENGE FOR SOTA LLMS?

We analyze the performance differences among categorizations that DL-Bench provides. Table 3 presents the pass@1 scores that each LLM achieves for generated code in each categorization that DL-Bench provides: stages in DL/ML pipeline, ML tasks, and input data types. Among all LLMs, the most advanced LLM, O3-Mini, consistently outperforms others in all categorizations. However, in object detection and recommendation, DeepSeek-V3 and Claude 3.5 perform better.

**Stages in pipeline:** Among stages in the DL/ML pipeline, *pre/post processing* generated code has the highest average pass@1 score of **31.9%**. Code in these stages varies significantly because it prepares and cleans the input and formats output data for various models. This makes samples of this type the most available in training data and could explain the higher pass@1 scores. On the other hand, LLMs struggle to generate code for the *model construction* stage, with the lowest average pass@1 score of **22.3%**. This is because the code for this stage is more complex, often longer, and project-specific.

> **Finding 3:** LLMs perform best (average pass@1 score of 31.9%) in pre/post processing stages and worst (average pass@1 score of 22.3%) in model construction. These differences could be due to the high availability of training data for pre/post processing stages, and the more complex and project-specific nature of code in model construction,

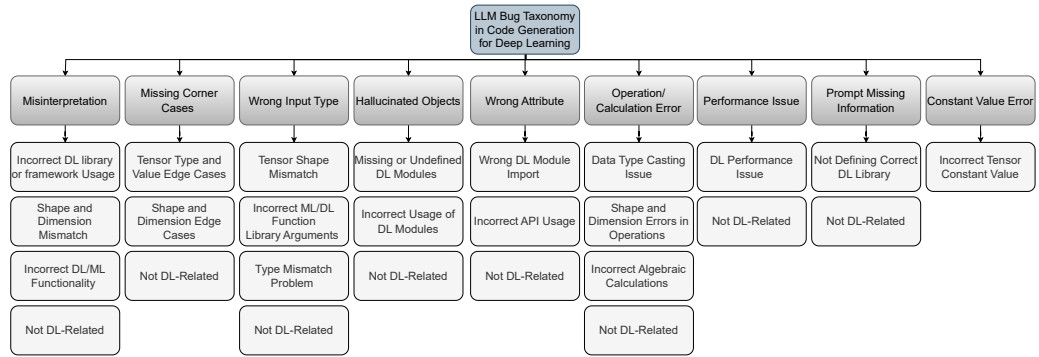

Figure 4: Taxonomy of bugs in DL generated code. (Only categories with DL-related subcategories).

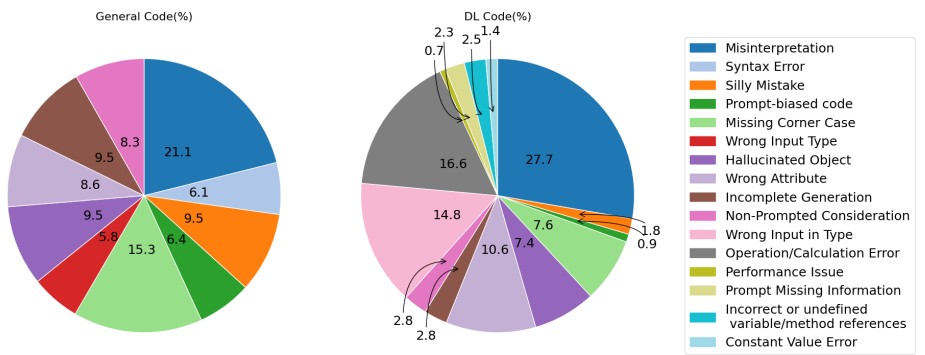

Figure 5: Distribution of bugs in general code vs. DL code generated by LLM

**ML tasks:** Table 3 presents a significant disparity in the pass@1 score of generated code across ML tasks. Notably, scores for the *recommendation* task are the highest (44.1% average), with the best score of 56.9% for Claude 3.5 Sonnet. On the other end of the scale, *object detection* and *image segmentation* tasks' scores are the lowest (averaged 18.4% and 19.8% respectively). These results indicate that each ML task type has its characteristics that LLMs can or cannot yet capture. Specifically, image processing code for object detection and image segmentation remains challenging.

> **Finding 4:** Different ML/DL tasks vary in complexity, affecting LLMs' code generation abilities with varying pass@1 scores averaged from 44.1% to 18.4%. Each LLM can have its strengths and weaknesses when generating code for different ML tasks.

**Input data types:** Across different types of input, the result in Table 3 indicates a more consistent pass@1 of all LLMs, except for textual data, where LLMs exhibit better performance (averaged 38.1%). We assume that most textual input data types are tokenized and converted before being processed in the DL model, which makes functions that deal directly with textual input data types quite standard and easier to generate. On the other hand, performance for image-related tasks perform the worst with averaged score of 24.9%. This can be attributed to the inherent complexity and lack of consistent structure in image data, such as varying shapes, resolutions, and channel configurations (e.g., grayscale vs. RGB).

> **Finding 5:** Among input data types, image data with more complex structures is the hardest to generate code for, with the lowest average pass@1 score of 24.9%. In contrast, textual data tasks achieved higher performance (average 38.1%), likely due to more deterministic coding in the pre-processing stages.

## 5.2 WHAT ARE THE COMMON BUGS IN GENERATED DL-SPECIFIC CODE?

To investigate this question, we build a taxonomy of common bug patterns and issues that arise in DL code generated by GPT-4o (the best model at the time of analysis). This taxonomy is an expansion of Tambon et.al Tambon et al. (2024)'s bug taxonomy for LLM-generated regular code. Following

Table 4: Distribution of stages in DL/ML pipeline for DS-1000 (predicted) and DL-Bench (actual).

| Stage in pipeline | Pre/Post Processing | Model Construction | Training | Inference | Evaluation |
|---|---|---|---|---|---|
| DL-Bench (actual) | 210(40.3%) | 119(22.8%) | 75(14.4%) | 59(11.3%) | 57(10.9%) |
| DS-1000 (predicted) | 932(93.2%) | 35(3.5%) | 14(1.4%) | 14(1.4%) | 5(0.5%) |

the same procedure as our labeling process, three authors manually investigate all GPT-4o failures and categorize them following Tambon et.al's taxonomy. At the same time, the annotators identify the DL-specific sub-categories for each failure. The result is the taxonomy presented in Fig 4. The appendix gives a detailed explanation of each bug type and sub-category.

**Differences in failures of the DL and general generated code:** Tambon et. alTambon et al. (2024) analyzed failures when CodeGen models generate code for the general tasks. Figures 5 show the distributions of the bug types when generating general code vs DL code. On the one hand, *misinterpretation* (purple) is a common bug when generating both general code and DL code; however, due to more complex logic and arithmetic requirements, LLMs more often make this mistake when generating DL code. On the other hand, since GPT4o is much more capable compared to CodeGen models used by prior work, errors such as *incomplete generation* (green), *silly mistake* (dark gray), and *syntax error* (yellow) occur at a much lower rate.

Additionally, we introduce several new categories of bugs that only arise in DL code generation. Firstly, *errors in arithmetic and logical operations*(light blue) occur when incorrect calculations or flawed logical code are generated. Secondly, *performance*(light brown) issues involve inefficiently generated code with slow execution times, excessive memory consumption, or suboptimal utilization of resources. Lastly, *prompt missing information*(light purple) occurs when the prompts are missing details to fully address the problem at hand, resulting in incomplete or partially implemented solutions. These new categories identify important challenges that are unique to DL code generation.

> **Finding 6:** *Misinterpretation* is a common issue in both generated general code and DL code; however, due to more complex logic and arithmetic requirements, LLMs are more likely to make this mistake when generating DL code. *Errors in arithmetic and logical operations*, *performance*, and *prompt missing information* emerged as new issues that are specific to DL code generation.

**Bugs in human-written compared to LLM-generated DL code:** Prior study Islam et al. (2019) has identified the most common types of bugs in human-written DL code (including logic errors, API misuse, and data-related issues), with API misuse and data flow bugs being the most prevalent issues in TensorFlow and Pytorch, respectively. Although API misuse remains a frequent issue in DL generated code, data structural problems, such as tensor mismatches and dimensional errors, occur more frequently. Human-written and LLM-generated DL code both often contain logic errors. This similarity may stem from the fact that LLMs are trained on human-written code, thereby inheriting logical structures and concepts from human programmers.

> **Finding 7:** Due to LLMs' weaknesses, LLM-generated DL code contains more data structural problems, such as tensor and dimension mismatches. However, due to reliance on human-generated training data, LLM-generated DL code shared bug patterns such as logic and API misuse errors.

## 6 DISCUSSION: DL-BENCH IN PRACTICE

One usage of the categorized data in DL-Bench is to train classifiers that can provide DL-specific categorization for other unlabeled datasets(e.g., DS-1000) to improve their quality. To test this potential usage, we train a BERT classifier to predict the *stage-in-pipeline* for each input prompt. The classifier uses the BERT tokenizer, BERT encoder, and a linear classifier. The optimization is performed with AdamW ($\eta = 2 \times 10^{-5}$, $B = 8$, $E = 10$), and five-fold cross-validation confirms stable generalization (average weighted $F_1 = 0.56 \pm 0.06$).

To verify the accuracy, we conducted manual labeling of 100 instances, which shows our classifier has a high accuracy of $95.0 \pm 5.3\%$ (with 99% confidence). This indicates a high level of generalization of DL-Bench categorization when applied to other DL-related benchmarks. Table 4 presents the predicted distribution for DS-1000 as well as the actual distribution for DL-Bench. The distribution differences further distinguish DL-Bench and DS-1000. Where DS-1000 mainly focuses on pre/post-processing, DL-Bench contains data from all stages of the DL/ML pipeline.

Table 5: Pass@1 rates for improved prompting techniques with DL-Bench's insight.

| Dataset | Prompting Technique | O3-Mini | DeepSeek-V3 | QwenCoder | GPT-4o | Claude 3.5 Sonnet | Llama 3.1 70b | Mixtral 8*22B | Avg. |
|---------|---------------------|---------|-------------|-----------|--------|-------------------|---------------|---------------|------|
| DS-1000 | Zero-Shot | 61.0 | 61.7 | 57.3 | 51.1 | 61.9 | 40.9 | 39.3 | 53.3 |
|  | Three-Shot | 50.2 | 62.6 | 57.5 | 54.2 | 64.8 | 51.4 | 42.0 | 54.6 |
|  | Stage-Predicted Three-Shot | 54.3 | 64.1 | 58.9 | 57.7 | 66.4 | 54.0 | 47.6 | 57.5 |
| DL-Bench | Zero-Shot | 35.1 | 30.5 | 30.2 | 30.5 | 26.7 | 23.9 | 22.8 | 28.5 |
|  | Three-Shot | 37.1 | 32.3 | 24.5 | 33.4 | 32.3 | 27.8 | 27.2 | 30.7 |
|  | Stage-Predicted Three-Shot | 38.2 | 34.1 | 26.3 | 35.2 | 33.6 | 29.6 | 28.1 | 32.2 |

> **Finding 8:** DL-Bench could complement other DL-related benchmarks by providing training data for categorization classification. Such a stage classifier can have a high accuracy ($95.0 \pm 5.3\%$ at 99% confidence when extending DS-1000).

Few-shot prompting emerged as a way to improve vanilla zero-shot prompting. However, guided shots from the same code category could potentially provide even more uplift in performance. To gauge the potential, we perform a preliminary experiment with three-shot prompting where the shots are random reference samples, or samples in the same DL stage as the question. Since the stage the prompt belongs to is not available, we use the previously described classifier to predict the stage. Table 5 shows the pass@1 rate for SOTA LLMs using the three prompting approaches on DL-Bench and DS-1000. For three-shot prompting, we perform the experiment twice and present the average pass@1 rates. Zero-shot, without any examples, performs the worst with an average pass@1 rate of 53.3% on DS-1000 and 28.5% on DL-Bench. By including three examples, three-shot prompting has a better average pass@1 rate of 54.6% on DS-1000 and 30.7% on DL-Bench. When providing shots for each prompt, we made sure that the shots do not overlap with the prompt. When each query is paired with snippets that belong to the same predicted stage of the DL pipeline, the pass@1 rate improves significantly. Averaged across models, stage-predicted three-shot prompting yields a **4.2%** and **3.7%** boost over zero-shot in DS-1000 and DL-Bench. This indicates the value of having lower granularity categorization in a dataset, which can enable more sophisticated prompting and fine-tuning techniques, which in turn provide uplift in LLMs' performance.

> **Finding 9:** Classifiers built using DL-Bench categorization data can provide targeted shots in few-shot prompting to improve code generation performance. Overall, stage-predicted three-shot yields up to **4.2%** and **3.7%** boost over zero-shot techniques in DS-1000 and DL-Bench respectively.

## 7 LIMITATIONS AND THREATS TO VALIDITY

Even with the temperature parameter set to zero, our experiments still utilized non-deterministic models. While a lower temperature reduces randomness, it does not fully eliminate variability in the models' outputs Ouyang et al. (2023); Song et al. (2024). Also, even if we used the commonly used pass@k metric to evaluate model performance, prior research Shiri Harzevili et al. (2024) shows that passing all test cases does not guarantee complete code correctness (e.g., in edge cases).
We sourced data from various repositories related to DL and AI, but did not include all possible repositories or tags. Expanding the dataset could capture a wider range of use cases and code patterns. Data labeling was performed by four annotators, achieving strong inter-rater reliability. Despite this, some labeling conflicts persisted and were addressed through discussions to reach a consensus.

## 8 CONCLUSION

In this paper, we introduce DL-Bench, a benchmark for deep learning tasks related to code generation. The dataset comprises 520 instances, gathered from the most starred and recently updated GitHub repositories. We categorize the data based on the pipeline stage, ML task, and input data type. Additionally, our quantitative analysis of the performance of four state-of-the-art LLMs on DL-Bench reveals that DL code generation is challenging and DL-Bench can provide more insight to help improve the generation process. Using our taxonomy of issues found in LLM-generated DL code, the qualitative analysis reveals the distinct challenges that LLMs face when generating DL code compared to general code as well as the similarities and differences between human-written and LLM-generated DL code. Our discussion shows potential usages of DL-Bench's categorization data outside of benchmarking usages. DL-Bench's data is available in our Kaggle repository.

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

# Appendix

Table of Contents:

## A    RELATED WORKS

**Code Generation Benchmarks for Data Science and ML/DL:** Several benchmarks have been developed to evaluate code generation models in the context of data science and ML/DL tasks. JuICe Agashe et al. (2019), PandasEval and NumPyEval Zan et al. (2022), and JuPyT5 Chandel et al. (2022) provide datasets from Jupyter notebooks or data science libraries, with a focus on realistic usage scenarios. However, most of these benchmarks rely on similarity-based metrics such as BLEU for evaluation, due to the lack of accompanying test cases. In contrast, DL-Bench includes multiple assert-based test cases for each entry, enabling more reliable evaluation via metrics like pass@1.

JuPyT5 introduces the DSP benchmark with 1119 pedagogically curated problems featuring markdown context, assert-based unit tests, and implicit data dependencies, making it suitable for evaluating notebook-based code generation. Similarly, CERT provides PandasEval and NumPyEval for structured, API-heavy data science tasks and shows performance gains by anonymizing user-defined elements. JuICe offers a large-scale dataset from Jupyter notebooks with manually curated test sets derived from nbgrader assignments, although its evaluation also depends on similarity metrics.

Shin et al. Shin et al. (2023) focus specifically on ML/DL tasks using JuICe samples but still evaluate with similarity scores, which do not reliably indicate functional correctness. In contrast, DL-Bench offers task-level test cases for each function, allowing more precise evaluation of LLM performance in ML/DL scenarios.

DS-1000 Lai et al. (2023) collects 1000 data science problems from StackOverflow, primarily focusing on tasks like data preprocessing or transformation, with the support of test cases. However, the tasks are often limited to isolated code snippets rather than complete function-level implementations. MLE-Bench Chan et al. (2024) captures ML engineering workflows in the context of Kaggle competitions, focusing on end-to-end pipelines but lacking fine-grained test-based evaluation. AICoderEval Xia et al. (2024) further abstracts the evaluation to workflow-level code generation, treating setup and implementation as a black-box output, which can obscure the model's capabilities at the component level.

**General Code Generation Benchmarks:** Benchmarks like HumanEval Chen et al. (2021), MBPP Austin et al. (2021), APPS Hendrycks et al. (2021), AiXBench Hao et al. (2022), MultiPL-E Cassano et al. (2022), Spider Yu et al. (2018), CoderEval Yu et al. (2024), and RepoEval Zhang et al. (2023) have been widely used to evaluate LLMs on general-purpose programming. These benchmarks span various tasks such as competitive programming, repository-level generation, multi-language support, and SQL query generation from natural language. However, they are primarily focused on general programming capabilities and do not capture the domain-specific challenges of ML/DL code.

**Distinctive Features of DL-Bench:** DL-Bench differs from prior work in several key ways. First, it focuses exclusively on ML and DL software development tasks, offering function-level prompts that reflect real needs in the ML pipeline. Second, it categorizes each function based on the ML pipeline stage (e.g., preprocessing, model training), task type (e.g., classification, regression), and input data type (e.g., image, text, tabular), offering a richer annotation scheme. Third, unlike most benchmarks, DL-Bench is sourced from real GitHub repositories, ensuring practical relevance, and each entry includes assert-based test cases, enabling robust and reproducible evaluation using metrics like pass@1.

Overall, DL-Bench complements existing benchmarks by providing a granular, test-driven, and ML/DL-focused dataset that enables more realistic evaluation of LLMs in domain-specific development scenarios.

# B  DATASET STATISTICS

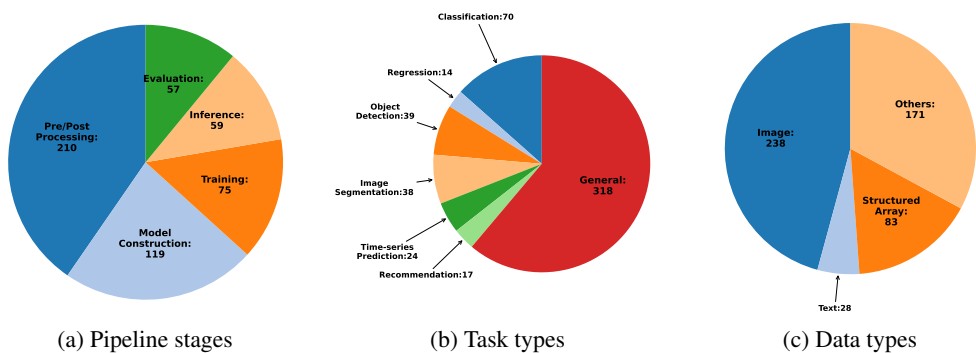

(a) Pipeline stages      (b) Task types      (c) Data types

Figure 6: Distribution of code samples in each category

DL-Bench consists of 520 instances of AI and DL data points (filtered from over 2,000 raw data points). The data is curated from 30 GitHub repositories (selected from an initial pool of 160 related repositories). To ensure an accurate evaluation of code generation techniques under test, each prompt instance in DL-Bench is accompanied by at least three test cases (six test cases on average). One of DL-Bench's contributions is the categories that we assign to each data point. As mentioned in Section 3, each data point is assigned a label for which stage of the ML pipeline it belongs to, a label for which ML task it helps solve, and a label for the type of input data. This information enables users of our benchmark to perform an in-depth analysis of their proposed technique with respect to multiple ML-specific aspects. We demonstrate this in our empirical study presented in Section 4 later.

> Write a Python function **draw_point2d** to set [x, y] coordinates in an image tensor (**grayscale** or **multi-channel**) to a given color, returning the **modified image**.

Figure 7: An example prompt for Pre/Post processing

> Create the `**__init__**` method for the **FCNN** class initializes a fully connected neural network with **input/output units**, **activation functions**, and **hidden layer sizes**. If not provided, default **hidden_units** to **(32, 32)**.

Figure 8: An example prompt for Model Construction

Fig 6 represents the distribution of DL-Bench's data in each categorization. In terms of the stages in the ML pipeline (Fig (a)), our dataset well covers the five stages of the ML pipeline with the pre/post-processing stage having the most (210) representative samples. Fig 7 lists the prompt to generate a pre/post-processing "draw_point2d" function that can be used to highlighting key points of interest in output images. The model construction stage contains the second-most (119) samples such as the one shown in Fig 8. This example shows the prompt to generate the "__init__" method for a fully connected neural network (FCNN). Other ML stages have an equal share of samples. This indicates a balanced dataset that covers all ML stages.

> Create a Python function **to_image** that accepts an input of type Union[torch.Tensor, PIL.Image.Image, np.ndarray] and returns a **tv_tensors**.Image. The function should check the input type and convert it accordingly

Figure 9: Example of General Task

Create a Python function **classification_metrics** that takes `ground_truth` and `retrieved` dictionaries and returns per class precision, recall, and F1 scores. Class 1 is assigned to duplicate file pairs while class 0 is for non-duplicate file pairs.

Figure 10: An example of Classification Task.

Most of our data serve more than one ML task type, hence 318 (over 61%) instances are labeled as *General* as shown in Fig (b). For example, Fig 9 shows **to_image** function handles data type conversions and pre-processing to standardize image inputs, without performing any specific machine learning task. However, for the cases that serve a specific ML task, our dataset covers all ML tasks evenly with 14 to 70 instances each. Among these, the classification task has the most representative of 70 data points. For example, Fig 10 shows a classification task, calculating precision, recall, and F1 scores for both duplicate and non-duplicate file pairs to evaluate the performance of a classification model. On the other hand, The regression task is not as popular with only 14 data points.

Image data is the most popular input data type with 238 instances (nearly 46%) as shown in Fig (c). In some cases where the input data to the function is missing or not the input to the model, we categorize them into the Others category which contains 171 instances. An example of such cases is presented in Fig 8, where the initialization method constructs a new neural network model, however, information on the input type of such networks is not available. Textual data has the least instances since most of the time, textual data is tokenized and presented as either a data array or general tensor.

## B.1 SEMANTIC DIVERSITY ANALYSIS BETWEEN DL-BENCH AND DS-1000 AND AICODEREVAL

We gathered tokens from influential DL and ML papers to capture the specialized terminology used in this field. Key sources include foundational works like Attention Is All You Need Vaswani et al. (2017), Deep Residual Learning forImage Recognition He et al. (2016), YOLOv4: Optimal Speed and Accuracy of Object Detection Bochkovskiy et al. (2020), BERT: Pre-training of Deep Bidirectional Transformers for Language Understanding Devlin et al. (2019), EfficientNet: Rethinking Model Scaling for Convolutional Neural Networks Tan & Le (2019), Learning Transferable Visual Models From Natural Language Supervision Radford et al. (2021), Neural Machine Translation by Jointly Learning to Align and Translate Bahdanau et al. (2014), and Sequence to Sequence Learning with Neural Networks Sutskever et al. (2014).

From these papers, we extracted common DL tokens from their GitHub repositories and source code, focusing on terms frequently used in DL models and architectures. Examples include:

- **Architecture Terms**: `cnn`, `rnn`, `transformer`, `lstm`, `gru`, `autoencoder`, `resnet`, `mobilenet`, `efficientnet`
- **Optimization**: `backpropagation`, `gradient_descent`, `adam`, `rmsprop`, `sgd`, `momentum`, `learning_rate`
- **Components**: `dropout`, `batchnorm`, `layernorm`, `relu`, `softmax`, `attention`, `dense_layer`, `conv2d`
- **Training**: `epoch`, `batch`, `overfitting`, `underfitting`, `weight_decay`, `cross_entropy`, `loss`
- **Processing**: `tokenizer`, `embedding`, `feature map`, `convolution`, `padding`, `pooling`, `strides`
- **Other Common Terms**: `activation`, `tensor`, `inference`, `regularization`, `initialization`, `hyperparameter`, `weight_matrix`

The list of all terms is available in our repository.

To quantify the presence of these DL/ML-specific terms, we computed a DL-relevance ratio for each instance in our benchmark datasets. Let $d_i$ denote the description of instance $i$, and $T(d_i)$ its preprocessed token set. The DL-relevance ratio $r_i$ is defined as:

$$r_i = \frac{|\{t \in T(d_i) \mid t \text{ contains a DL/ML keyword}\}|}{|T(d_i)|}$$

This metric captures the proportion of DL/ML-related tokens in each textual description, providing a quantitative measure of the DL specificity of each benchmark. For example, a code snippet containing terms like `conv2d`, `batchnorm`, `softmax`, and `dropout` would have a higher relevance ratio than one primarily focused on generic algorithmic operations.

## C    DETAILED BENCHMARK CONSTRUCTION PROCEDURE

### C.1    RAW DATA EXTRACTION

This phase consists of six semi-automatic steps that crawl data from GitHub repositories to generate a list of function definitions and their test cases.

**Repository Selection:** We curated our data from the top 1000 starred DL-related GitHub repositories to include high-quality and widely used DL-related functions.

In step ①, we filtered GitHub projects with one of 30 DL-related tags such as "neural-networks", "pytorch", and "computer-vision" (we provided the complete list of tags in our repository). Specifically, we select the tags by collecting from DL and AI-related GitHub repositories and filtering the most relevant ones to get the final 30. In step ②, we select 160 most relevant projects for DL-Bench and retain only projects that: 1) are DL related (i.e., use DL libraries, or perform DL tasks like segmentation or detection), 2) have sufficient test cases (averaging at least three per function), and 3) include thorough documentation, such as source code docstrings or README files.

**Function Extraction:** One of the main design choices of DL-Bench is to include a set of reliable and robust test cases for each benchmark entry. This is because programming languages are different from natural languages. Specifically, generated code can fulfill all of the functional requirements but could have a low BLEU score when compared with the ground truth codeTran et al. (2019). This means that using text similarity metrics such as BLEU score as evaluation metrics is not the best method to evaluate code generation techniques. Instead, test cases (functional and non-functional) passing rate should be used to reliably access a new code generation approach.

In step ③, we crawled selected repositories for test files using standard test file name patterns such as tests/test_file_name.py Madeja et al. (2021). In step ④, for each test file, we extract test cases using common patterns in Python test suites, such as the @pytest decorator. Once we identified all test cases, in step ⑤, we performed call graph analysis to track and collect all functions under test (excluding third-party function calls). The definitions of each of those functions are then extracted in step ⑥ to form the bases for our ground-truth code samples.

### C.2    LABELING PROCEDURE

The labeling procedure involves three semi-automatic steps to generate and refine a prompt and assign categorizations for each entry in our DL-Bench dataset. To determine the best procedure and criteria for our manual process, we perform a small trial run of the manual process on a small sample of the data points. In this trial run, we ask each reviewer to provide feedback on the labeling criteria so that when we start our full run we have the most comprehensive and accurate manual process possible.

**Prompt Generation:** In step ⑦, we utilize two sources of data to create the code generation prompts: 1) the doc-strings provided by developers, which describe the functionality and parameters of the code, and 2) the function definitions themselves, which can be used to generate candidate prompts. Specifically, We take advantage of the function definitions to explain the code, and by combining them with their respective doc-strings (when available), we generate the initial candidate prompt by querying GPT-4o with the template as described in Fig. 2.

However, generated prompts require manual validation to ensure accuracy and relevance. This review process is essential to refine prompts and guarantee quality for subsequent use Shrivastava et al. (2023). We further refine prompts based on the following criteria: (1) contain clear, sufficient information for code generation, (2) specify input and output format, and (3) cover error handling and boundary conditions. More details are in the appendix.

If the prompt does not meet the mentioned criteria, the annotators propose and agree on changes that bring it up to the expected quality. This reviewing process produces prompts that are not only technically correct but also include details essential to code generation.

Our manually refining process of generated prompts mitigates the risk of data leakage since. This process creates original natural language prompts which have not been previously exposed to any new language model.

**Data Filtering and Validation:** After compiling all the data (i.e., the ground truth, test cases, and candidate prompts), in step ⑧, we manually evaluate each function meticulously, reading and modifying the prompts following a set of criteria. Specifically, we discard general codes (e.g., those

for reading text files) that are not DL related. In this step, the annotators independently assess the prompt's clarity, relevance to DL-related tasks, and overall usability with the following criteria: (1) serving key DL tasks, (2) utilization of popular DL frameworks, and (3) algorithms' relevancy and clarity.

**Labeling:** In step ⑨, we assign labels for each data point based on the role of the function in the ML pipeline (e.g., pre/post-processing, model construction), the ML tasks (e.g., classification, regression) it solves, and types of data (e.g., image, text) it operates on. For each data point, three co-authors thoroughly analyze and assign appropriate labels. We use a majority vote to finalize the labels and modify the prompts accordingly. Specifically, we assign the following labels when appropriate to each data point: Stage in the ML pipeline, ML task type, and Input data type.

Once each reviewer completes their assessments, the team meets to discuss any discrepancies and reach a consensus on the final labels. Due to our detailed instructions and guidelines, we achieve a high inter-rater reliability of 0.83 measured by Krippendorff's alpha Zapf et al. (2016)(measures of more than 0.8 indicating strong agreement.

The labeled data is carefully documented, including notes on the decision-making process for transparency and future reference. Instances are organized, with labels to ensure easy retrieval and analysis in later stages of research. To enable easier benchmark utilization (i.e., running test cases), the relevant projects are set up in virtual environments along with appropriate dependencies and ready-to-run testing scripts.

This rigorous review and labeling process ensures that each instance in the dataset is not only relevant and useful but also thoroughly understood and appropriately categorized, contributing to a robust and reliable benchmark.

## D    CANDIDATE PROMPT FILTERING CRITERIA

In this appendix, we describe the criteria of filtering and refining prompts to ensure clarity and completeness.

**Contains clear sufficient information for the code to be generated**  This assessment aims to ensure the prompt's clarity and comprehensibility for a human expert. Annotators check that the prompt includes all essential variables, functions, and definitions for high-quality code generation, providing enough information to clearly explain the problem. The human expert serves as the benchmark to set a high standard for future code generation. We also verify that the prompt provides sufficient guidance, including specific coding conventions or required components.

**Specifies the input and output format**  Since our test cases require certain input and output formats, it is important to check such details in the candidate prompt to enable our test cases to function correctlySahoo et al. (2024); Chen & Moscholios (2024). In other words, without precise definitions of the input and output specifications, the generated code might not align with the expected test parameters, resulting in false negative results during evaluation. Error and exception handling are also considered in this question. For example, we specifically check whether the prompt accounts for handling cases such as "ValueError", "TypeError", or other domain-specific exceptions that the function might raise. This will ensure that the code will be correctly evaluated given our extracted test cases.

**Covers error handling and boundary conditions**  Similar to input and output specification, error handling and boundary conditions are often part of the required testing parameters By ensuring that the prompt includes such details, we ensure that the passing rate truly reflects the performance of the code generation under test.

## E    FINAL DATA FILTERING AND VALIDATION CRITERIA

This appendix outlines the criteria used to filter and validate data, ensuring alignment with key DL tasks, proper use of AI frameworks, and clarity in algorithm implementation.

**Serving key DL tasks**  The prompt and the associated function should be closely aligned with significant DL tasks such as image recognition, regression, item recommendation, object detection, label prediction, and natural language processing tasks. This criterion ensures that our dataset contains all important and relevant data pointsXie (2024).

**Utilization of popular DL frameworks**  The code should efficiently use widely recognized AI frameworks (when appropriate), such as TensorFlow, PyTorch, or Keras. This criterion

ensures our dataset represents typical DL code with a heavy emphasis on reusabilityAssi et al. (2024).

**Algorithms' relevancy and clarity** The code should implement DL-specific algorithms (e.g., edge detection algorithms, Principal component analysis, or Stochastic gradient descent). The code should also be well-documented and easy to understand. Complex algorithms must strike a balance between technical depth and clarity to ensure usability.

## F DATA CATEGORIES AND LABELS

In this appendix, we provide details of three key sample categorizations: the stage in the ML pipeline, the ML task type, and the input data type.

### F.1 STAGE IN THE ML PIPELINE

This label indicates the stage that the code is in within the ML pipeline: *Pre/post Processing*, *Model Construction*, *Training*, *Inference*, or *Evaluation & Metrics*. The annotators determine whether the function is related to a stage by analyzing the code and comment to find information that is related to the specific stage. For example, code that specifies a convolutional neural network (CNN) architecture with layers such as convolutions or pooling would fall under the Model Construction category.

**Pre/Post Processing** Code in the pre or post-processing stage often manipulates data (input or output). For example, pre-processing code cleans or augments input data, whereas post-processing code augments output data for visualization. Due to the ambiguity at the function level, we have a combined category for pre and post-processing codeWen et al. (2020).

**Model Construction** This stage defines the network architecture and sets up the computational graph for deep learning models, including defining layers, activation functions, and layer connections. Examples include defining CNN architectures and forward pass logic. Loss functions are part of this stage, but optimization steps are in the training phaseHoward et al. (2019).

**Training** The training stage optimizes the model's parameters using a loss function and optimization algorithm. This includes backpropagation and weight updates. Code for gradient descent using optimizers like Adam or SGD and looping over epochs and batches falls under this stageDiederik (2014).

**Inference** Inference code is used to generate labels based on a trained model. It processes new input data and outputs results, such as classifications or detections, without changing model parameters. This stage emphasizes speed and efficiency for real-time deploymentKirillov et al. (2019).

**Evaluation & Metrics** Code in this stage assesses the performance of a trained model using various metrics. It involves running the model on a validation/test dataset and comparing predictions to ground truth labels to measure accuracy, precision, recall, F1-score, etc.Wu et al. (2020).

### F.2 ML TASK TYPE

This label indicates the ML taskSarker (2021); Vinodkumar et al. (2023); Manakitsa et al. (2024) that the code is serving when applicable. The annotators examine the code to determine the type of task being solved, such as *Time series Prediction*, *Recommendation*, *Image Segmentation*, *Object Detection*, *Regression*, *Classification*, or *General*. Specifically, the annotators look for patterns in the code corresponding to each task. For instance, code that outputs bounding boxes and class labels for objects falls under the Object Detection category. In cases where the code can be used for multiple ML tasks (i.e., does not exclusively belong to a specific ML task), we assigned a *General* label.

**Classification** Classification tasks involve assigning input data to categories or classes. For example, models using softmax activation in the final layer for outputs like "dog" or "cat" fall under this category. Categorical cross-entropy loss is a common indicator.

**Regression** Regression tasks predict continuous values. Code indicating regression tasks often has linear activation functions in the final layer.

**Object Detection** Detection tasks identify objects and their locations within images. Code that outputs bounding boxes and class labels (e.g., YOLO, Faster R-CNN) and employs anchor boxes or non-maximum suppression is indicative of detection tasks.

**Image Segmentation** Segmentation tasks assign labels to each pixel in an image. Code involving semantic or instance segmentation (e.g., U-Net, Mask R-CNN) where the output is a mask with pixel-level classifications is a common example.

**Time Series Prediction** These tasks forecast future values using historical data. Code involving recurrent neural networks (RNNs), LSTM, GRU models, and loss functions like mean absolute error (MAE) or MSE is typical.

**Recommendation** Recommendation tasks suggest items or actions based on user data. Code implementing collaborative or content-based filtering algorithms, matrix factorization, or deep learning-based models for recommendations falls into this category.

**General** Code that is versatile and applicable to multiple ML tasks without being exclusive to a specific one is labeled as **General**.

F.3    INPUT DATA TYPE

This label indicates the input data type of the function. We focus on typical ML input data types such as *Image*, *Text*, *Structured Array* (i.e., tabular), and *Others*. The annotators analyze the processing flow of data to assign accurate labels. For example, techniques like flipping, cropping, or adding noise process image input. When the input data does not fit one of the typical types (image, text, structured array), we assign the Others label.

- **Image**—Processing for image data includes steps like resizing, normalization, and data augmentation. Code that resizes images (e.g., 224×224 for CNNs), normalizes pixel values, or applies augmentations (flipping, cropping, noise addition) typically signals image dataKrizhevsky et al. (2012).

- **Text**—Text processing involves tokenization, n-gram generation, stemming, lemmatization, and embeddings. Code that handles these processes and converts text into vectors (e.g., using TF-IDF, Word2Vec, BERT) indicates text dataLiu & Zhang (2018).

- **Structured Array**—Tabular data, where rows represent data points and columns represent features, is processed by normalization, one-hot encoding, or handling missing values. Code that reads CSVs into DataFrames and applies these techniques indicates structured array data, commonly used in regression or classification tasksChen & Guestrin (2016).

- **Others**—When input data does not match typical types (image, text, structured array), it is labeled as **Others**. This includes input such as model parameters or hyperparameters. For example, `def __init__(self, weight, bias=None)` initializing model components without direct input data processing falls under this label.

## G    LLM BUG TYPES AND DL-SPECIFIC SUBTYPES

In this appendix, we provide details for the common types of errors in LLM-generated code as well as our DL-specific subtypes.

**Misinterpretation:** *Generated code deviates from the prompt intention* The produced solution does not fulfill the user's original requirements or strays from the specified goals. This often indicates that the LLM has misunderstood or incompletely parsed the prompt.

**Incorrect DL Library or Framework Usage:** The generated code does not match the requested library or framework. For example, if the prompt asks for a TensorFlow implementation of a CNN, but the LLM generates the model using PyTorch instead, or if a user requests a NumPy-based neural network operation but the output code uses TensorFlow functions.

**Shape and Dimension Mismatch:** The LLM produces code with incorrect tensor dimensions that do not follow the prompt specifications. For example, if the prompt requests a fully connected layer expecting an input of shape $(64, 128)$, but the generated code initializes it with an input shape of $(128, 64)$, leading to a mismatch in matrix operations.

**Incorrect DL/ML Functionality:** The generated code does not implement the correct functionality as described in the prompt. For instance, if the prompt asks for a binary classification model using a sigmoid activation function, but the output code instead applies a softmax activation function intended for multi-class classification, altering the intended behavior.

**Syntax Error:** *Missing parenthesis, semicolon, or other syntax issues* Straightforward syntactic mistakes such as unclosed quotes, unmatched braces, or misplaced punctuation prevent the code from compiling or running properly.

**Silly Mistake:** *Redundant conditions, unnecessary casting* Simple but avoidable errors, such as repeating the same condition twice or performing extra type conversions with no purpose. While these do not always break the code, they reduce readability and hint at confusion in the model's reasoning.

**Prompt-biased Code:** *Code overly relies on examples from the prompt* The LLM anchors too strongly to the examples provided in the prompt, resulting in a solution that works only for the specific inputs shown rather than generalizing the logic for broader applicability.

**Missing Corner Cases:** *Edge cases not handled* The generated solution neglects special scenarios such as empty inputs, boundary values, or invalid parameters, leading to unreliable behavior outside of typical inputs.

> **Tensor Type and Value Edge Cases:** These bugs occur when operations fail due to unexpected tensor types or values. For example, using a tensor with `float32` data type in a function that expects integers or encountering issues when dividing by zero in a tensor.

> **Shape and Dimension Edge Cases:** Bugs of this type happen when operations fail because of unexpected edge-case shapes. For example, trying to perform a convolution on a tensor with a batch size of $0$ or a single dimension, such as $(1, 28, 28)$, when a shape like $(32, 28, 28)$ is expected.

**Wrong Input Type:** *Incorrect input type in function calls* The code passes incompatible data types to functions or methods (e.g., providing a string instead of a list), which causes runtime failures or nonsensical outputs.

> **Tensor Shape Mismatch:** The generated code provides tensors with incorrect shapes to functions, leading to shape-related errors. For example, passing a 3D tensor of shape $(batch, height, width)$ to a function that expects a 4D tensor of shape $(batch, channels, height, width)$, causing a runtime error in deep learning frameworks like PyTorch or TensorFlow.

> **Incorrect ML/DL Function Library Arguments:** These occur when invalid arguments are passed to functions. For instance, using `stride=-1` in a convolution function, which is not logically or mathematically valid.

> **Type Mismatch Problem:** The generated code uses tensors with incompatible data types in operations. For example, passing a tensor with data type `float32` to a function that expects `int64`, or attempting to index a tensor with a floating-point value instead of an integer, leading to type-related execution failures.

**Hallucinated Object:** *Nonexistent or undefined objects used* The LLM invents objects, classes, or modules that do not exist or have not been imported or defined. These errors result in runtime failures or developer confusion.

> **Missing or Undefined DL Modules:** This happens when a model, layer, or module that hasn't been properly defined or initialized is used. For example, attempting to forward-pass input through a neural network layer that hasn't been added to the model.

> **Incorrect Usage of DL Modules:** The generated code references deep learning modules, functions, or classes that do not exist or belong to the wrong framework. For example, calling `torch.nn.Dense()` instead of `torch.nn.Linear()`, or attempting to use `tensorflow.layers.Conv2D` instead of `tf.keras.layers.Conv2D`. These hallucinated module names cause import errors or incorrect function calls.

**Wrong Attribute:** *Incorrect/nonexistent attributes for objects or modules* The LLM references valid objects but assigns them invalid or incorrect attributes. These subtle errors often result from misunderstandings of library APIs or typos in the generated code.

> **Wrong DL Module Import:** Bugs of this nature arise when modules are imported incorrectly. For example, importing `jax` functions when the rest of the code is written in PyTorch, leading to incompatibilities during execution.

**Incorrect API Usage:** These bugs occur when a library API function is called incorrectly. For example, using the `train()` method instead of `fit()` for a Keras model or passing parameters in the wrong order to an optimizer.

**Non-Prompted Consideration:** *Non-requested features added* The LLM includes functionality unrelated to the requirements, often due to extraneous training data or contextual noise. This bloats the code and complicates its scope.

**Operation/Calculation Error:** *Errors in arithmetic or logical operations* The LLM makes errors in mathematical calculations or logical expressions, such as confusing addition with subtraction or mixing up operator precedence. These subtle mistakes produce incorrect results.

**Data Type Casting Issues:** These bugs occur when tensors or variables are cast into incompatible data types. For instance, casting a `float32` tensor into `int32` without considering the loss of precision, which may disrupt training.

**Shape and Dimension Error in Operations:** The generated code performs mathematical operations on tensors with incompatible shapes or dimensions, leading to incorrect computations or runtime failures. For example, attempting to add two tensors of shapes $(32, 64)$ and $(64, 32)$ without proper broadcasting, or performing a matrix multiplication between tensors with mismatched inner dimensions, such as $(4, 3) \times (5, 4)$, causing a shape misalignment error.

**Incorrect Algebraic Calculation:** These bugs refer to mathematical errors in computations. For instance, incorrectly normalizing data by dividing by the mean instead of the standard deviation, leading to improper scaling of input features.

**Performance Issue:** This category includes inefficiencies in the generated code that impact runtime or resource usage. Examples include unnecessary nested loops, unoptimized algorithms, or excessive use of memory. While the code may produce correct results, its suboptimal implementation can make it impractical for large datasets or real-time applications. Performance issues often arise because the LLM generates a brute-force solution without understanding optimization principles.

**DL Performance Issues:** These bugs refer to inefficiencies in implementation that degrade model performance. For instance, not using GPU acceleration for operations or improper batching strategies leads to high memory consumption and slow training.

**Prompt Missing Information:** *Incomplete or unclear prompts* The bug arises due to insufficient detail or ambiguity in the input prompt, leading the LLM to make assumptions or guess certain details when generating the code. For example, if the prompt does not specify edge case handling or input constraints, the model may overlook these aspects entirely. This highlights the importance of crafting precise and comprehensive prompts when using LLMs for code generation.

**Not Defining the Correct DL Library in the Prompt:** This occurs when the prompt or instructions fail to specify the appropriate library or framework. For example, a user asks a language model to generate PyTorch code but does not explicitly state this, leading to TensorFlow code generation instead.

**Incorrect or Undefined Variable/Method References** : **Variables or methods that are not defined or incorrectly referenced** The LLM generates code that includes references to variables or methods that do not exist or are improperly used, leading to runtime errors such as NameError or AttributeError.

**Constant Value Error:** *Incorrect constant value assignment* The LLM assigns incorrect or miscalculated constant values, such as setting a time-out period to `10ms` instead of `1000ms`, leading to unexpected behavior.

**Incorrect Tensor Constant Value:** This type of bug arises when tensors are initialized with incorrect values, leading to flawed model behavior. For example, initializing weights or biases with all zeros instead of random values causes issues in training dynamics.

## H   DISTRIBUTION OF FAILURES IN GENERATED DL CODE

Table 6 presents the distribution of bugs in LLM-generated DL code. The most prevalent issue is **deviation from the prompt**, accounting for the largest portion of errors. Unlike general LLM-generated code, DL code is more prone to **arithmetic and logical errors**, reflecting the complexity of numerical computations. Additionally, **incorrect input types in function calls** represent a significant share of the identified bugs, highlighting a common source of failures in generated DL code.

Table 6: Distribution of bugs in LLM generated code for deep learning

| Category | DL Related Categories | # of Occurances | |
|---|---|---|---|
| **Misinterpretation**: Generated code deviates from prompt intention | Incorrect DL library or framework Usage | 10 | |
| | Shape and dimension mismatch | 45 | |
| | Incorrect DL/ML Functionality | 13 | 120 |
| | Not DL-related | 52 | |
| **Syntax Error**: Missing parenthesis, semicolon, or other syntax issues | | | 0 |
| **Silly Mistake**: Redundant conditions, unnecessary casting | Not DL-related | 8 | 8 |
| **Prompt biased Code**: Code overly relies on examples from the prompt | Not DL-related | 4 | 4 |
| **Missing Corner Case**: Edge cases not handled | Tensor Type and Value Edge Cases | 8 | |
| | Shape and Dimension Edge Cases | 15 | 33 |
| | Not DL-related | 10 | |
| **Wrong input type**:Incorrect input type in function calls | Tensor shape mismatch | 3 | |
| | Incorrect ML/DL function library arguments | 16 | 64 |
| | Type mismatch problem | 23 | |
| | Not DL-related | 22 | |
| **Hallucinated Objects**: Nonexistent or undefined objects used | Missing or Undefined DL Modules | 9 | |
| | Incorrect Usage of DL Modules | 12 | 32 |
| | Not DL-related | 11 | |
| **Wrong Attribute**: Incorrect/nonexistent attributes for objects or modules | Wrong DL Module import | 8 | |
| | Incorrect API Usage | 17 | 46 |
| | Not DL-related | 21 | |
| **Non-Prompted Consideration**:Non-requested features added | Not DL-related | 12 | 12 |
| **Operation/Calculation Error**:Errors in arithmetic or logical operations | Data Type Casting Issues | 5 | |
| | Shape and Dimension Errors in Operations | 28 | |
| | Incorrect Algebraic Calculations | 18 | 72 |
| | Not DL-related | 21 | |
| **Performance Issue**: Poor Performance | DL performance issue | 2 | |
| | Not DL-related | 1 | 3 |
| **Prompt missing information**:Incomplete or unclear prompts | Not defining correct dl library | 4 | |
| | Not DL-related | 6 | 10 |
| **Incorrect or undefined variable/method references**: Variables or methods that are not defined or incorrectly referenced | Not DL-related | 11 | 11 |
| **Constant Value Error**:Incorrect constant value assignment | Incorrect Tensor Constant Value | 6 | 6 |

### H.1   SOME EXAMPLES OF INCORRECT LLM-GENERATED DL CODE:

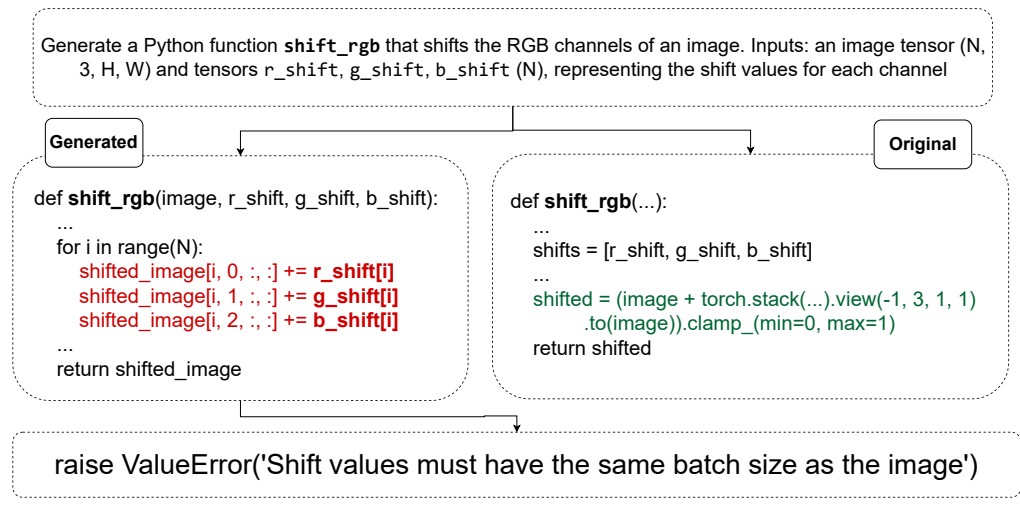

Figure 11: Mismatching data shapes: shifting variables need to be broadcasted to the image shape

**Example 1:** Figure 11 highlights an instance of dimensional mismatches in LLM-generated DL code. In this case, GPT-4o incorrectly assumes that each shift value can be applied directly to all pixels in the image channel, causing a shape mismatch.

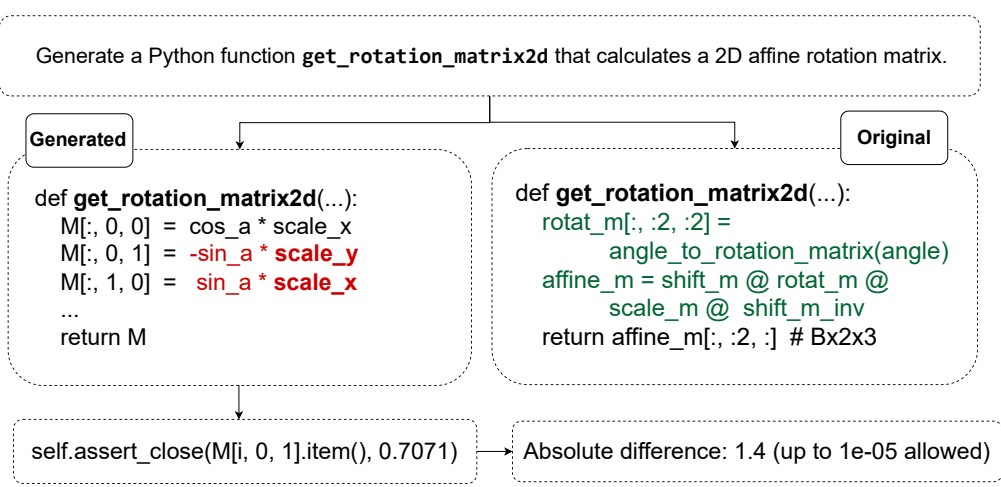

Figure 12: Incorrect processing of parameters: The axes scales need to be applied to both sin and cos

**Example 2:** An example of such logic-related bugs is shown in Figure 12, demonstrating how LLMs replicate logical reasoning errors that occur in human-written code. Here, GPT-4o applies $scale\_x$ only to the cosine, whereas the scaling factors $scale\_x$ and $scale\_y$ should be applied uniformly to both the sine and cosine components of the rotation matrix. This results in improper scaling along the axes and triggers a test failure.

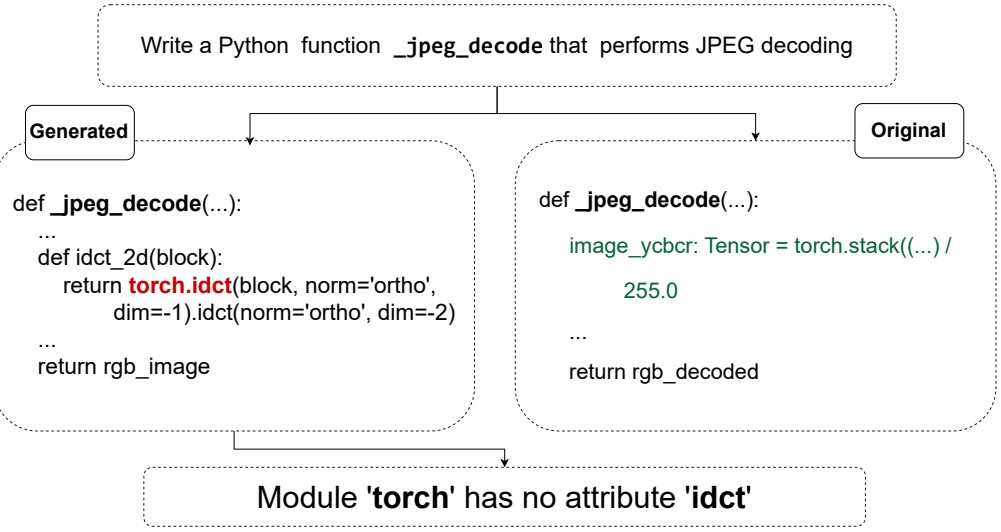

Figure 13: Wrong usage of a third-party library.

**Example 3:** Figure 13 provides an example of API misuse in LLM-generated code where GPT-4o attempts to call *torch.idct*, which is not implemented in PyTorch. One possible fix is to provide more context concerning third-party libraries. For example, one could hint to LLMs to use *scipy* instead, resulting in *scipy.fftpack.idct(x.numpy(), norm=norm)* instead.

## I  MORE RESULTS

This appendix provides a more extensive quantitative analysis of model performance on DL-Bench, expanding on the summary presented in the main text. In particular, we report complete pass@k

Table 7: Pass@k (%) on DL-Bench at temperature $= 0.3$.

| Model | Pass@1 | Pass@3 | Pass@5 |
|---|---|---|---|
| O3-mini | 36.7 | 38.1 | 40.2 |
| DeepSeek V3 | 31.7 | 34.6 | 36.7 |
| GPT-4o | 30.5 | 33.9 | 37.9 |
| Claude 3.5 Sonnet | 30.3 | 32.5 | 35.6 |
| LLaMA 3.1 70B | 27.8 | 29.4 | 33.9 |
| Mistral 8×22B | 26.4 | 27.3 | 31.4 |
| Qwen Coder 2.5 | 24.1 | 26.1 | 29.0 |

Table 8: Variance across five runs on DL-Bench (temperature $= 0$). Lower variance indicates more stable performance.

| Run / Stat | O3-Mini | DeepSeek V3 | GPT-4o | Claude 3.5 Sonnet | LLaMA 3.1 70B | Mistral 8×22B | Qwen Coder 2.5 |
|---|---|---|---|---|---|---|---|
| Run 1 | 36.9 | 31.4 | 31.2 | 30.3 | 27.5 | 26.1 | 23.4 |
| Run 2 | 35.6 | 30.6 | 29.4 | 29.8 | 27.1 | 23.9 | 21.7 |
| Run 3 | 34.6 | 28.5 | 28.3 | 29.9 | 25.8 | 24.0 | 22.7 |
| Run 4 | 34.6 | 32.1 | 30.8 | 31.2 | 28.3 | 22.5 | 23.8 |
| Run 5 | 33.7 | 30.2 | 31.6 | 31.4 | 25.2 | 23.1 | 22.5 |
| AVG | 35.1 | 30.5 | 30.2 | 30.5 | 26.7 | 23.9 | 22.8 |
| VAR | 1.49 | 1.86 | 1.89 | 0.55 | 1.60 | 1.86 | 0.67 |
| STD | 1.22 | 1.36 | 1.37 | 0.74 | 1.26 | 1.36 | 0.82 |

statistics and examine how allowing multiple generation attempts influences the success rate of each evaluated LLM.

### I.1 PASS@3 AND PASS@5 PERFORMANCE ON DL-BENCH

Table 7 presents the exact **pass@3** and **pass@5** scores of the seven representative LLMs when decoding with temperature 0.3. These results reveal the extent to which each model benefits from additional generation attempts. O3-Mini achieves the highest success rates with **38.1% pass@3** and **40.2% pass@5**, gaining about two percentage points when moving from three to five attempts. DeepSeek-V3 follows closely at **34.6%** and **36.7%**, while GPT-4o records **33.9%** and **37.9%**, representing the largest improvement (approximately four percentage points) among all models. Claude 3.5 Sonnet reaches **32.5%** and **35.6%**, and LLaMA 3.1 70B attains **29.4%** and **33.9%**. Among the smaller open-weight baselines, Mistral 8×22B achieves **27.3%** and **31.4%**, while Qwen Coder 2.5 delivers the lowest performance with **26.1%** and **29.0%**. Across all models, the absolute improvements from pass@3 to pass@5 remain relatively limited—generally within 2–4 percentage points—indicating that even with multiple generation attempts, current state-of-the-art LLMs continue to face considerable difficulty in producing fully correct ML/DL-specific code on DL-Bench. This further highlights the benchmark's effectiveness in exposing the limitations of modern code generation systems beyond what existing datasets such as DS-1000 can capture.

### I.2 VARIANCE OF DIFFERENT RUNS

Table 8 reports the exact pass@1 scores for each of the five independent runs together with the computed mean (AVG), variance (VAR), and standard deviation (STD). O3-Mini consistently achieves the highest average pass@1 score (**35.1%**) with a variance of **1.49** and standard deviation of **1.22**. DeepSeek V3 and GPT-4o show slightly higher variability (variance **1.86** and **1.89**, respectively) but still maintain mean scores around **30%**. Claude 3.5 Sonnet is the most stable, with a variance of only **0.55** (standard deviation **0.74**) around its **30.5%** mean. LLaMA 3.1 70B exhibits a variance of **1.60**, Mistral 8×22B also **1.86**, and Qwen Coder 2.5 remains relatively steady with a variance of **0.67**. These results show that even the lowest-performing models provide reproducible outcomes across repeated evaluations, reinforcing the robustness of the comparative analysis in the main text.

### I.3 RESULTS BASED ON TIMELINE

Table 2 shows the exact pass@1 scores on DL-Bench when tasks are filtered by their publication date relative to the October 2023 cutoff. These results reveal how each model's accuracy shifts as only the most recent tasks are considered. O3-Mini achieves the highest overall score at 35.1%, but its accuracy drops to 32.8% after January 2024, declines further to 29.6% after May 2024, and

then rises slightly to 30.5% after September 2024. DeepSeek V3 decreases from 30.5% overall to 27.5% after September 2024, while GPT-4o falls from 30.2% to 25.7% in the same period. Claude 3.5 Sonnet shows a more moderate decline from 30.5% to 27.6%. Among the open-weight models, LLaMA 3.1 70B drops from 26.7% to 25.0%, and Mistral 8 × 22B goes from 23.9% to 23.1%. Qwen Coder 2.5 remains comparatively low but stable, varying only between 22.8% and 24.2%. Overall, the consistent downward trend across most models highlights how the live version of DL-Bench continually surfaces fresh, previously unseen challenges that cannot be solved simply by exploiting prior training data, underscoring the benchmark's value for continual evaluation of LLMs on emerging ML/DL code-generation tasks.

