# OpenReview forum: "DL-Bench: Deep learning specific code generation benchmark"
_ICLR.cc/2026/Conference — Submitted to ICLR 2026_

### Official Review · Reviewer_GnjX · 2025-10-29

**Soundness:** 2
**Presentation:** 3
**Contribution:** 2
**Rating:** 2
**Confidence:** 4

**Summary:**

This paper introduces a new benchmark, DL-Bench, that evaluates LLMs’ ability to generate deep-learning-specific functions, covering deep-learning (DL) problems among three aspects: different phases, different tasks and different input data types. The main contribution is the benchmark itself.
This benchmark is novel (different from previous benchmarks) on multiple dimensions: (1) it occupies a semantically distinct region in the embedding space, (2) it contains more DL/ML specific keywords, (3) the oracle solutions are more complex, and (4) the pass@1 on DL-Bench of the SOTA LLMs is significantly lower than the pass@1 on previous benchmarks.
Analysis on DL-Bench provides new insights: (1) Analysis on LLMs’ generated code on DL-Bench find different bug distribution from that on general code. (2) Performance difference across different categories indicates insights for potential improvement. (3) When used as examples in few-shot prompting, DL-Bench can improve the performance of the problems in the same category.

**Strengths:**

This paper provides the various experiments from different aspects on its differences from previous benchmarks, and performs more analysis on LLMs’ generated code on DL-Bench. The paper clearly describes the data collection process and all the prerequisites of the analysis experiments. DL-Bench covers 520 samples from different DL libraries, constituting a broad context to examine LLMs’ ability to generate DL-specific functions.

**Weaknesses:**

The evidence in the paper only shows the difficulty (pass@1 on this benchmark is lower, the oracle solution is more complex) and the semantic difference (semantically distinct in embedding space), but none of them shows the quality of this benchmark. A more difficult and different benchmark is not automatically a better benchmark.

Major questions/limitations:
The quality of DL-Bench is questionable. Many experiments only show that DL-Bench is different from/more difficult than existing DL benchmarks, but they do not guarantee the quality of DL-Bench. On the other hand, from many evidences, I believe the quality of DL-Bench is questionable.
  1. Prompt: One of the bug categories in this benchmark is “prompt missing information” (Section 5.2, Finding 6), but this category only means the quality of the prompt provided by DL-Bench is not good enough.
  2. Function signature: The prompt does not always include the signature of the function to be generated. For example, the `load_model` function in the dataset does not contain the exact function signature, but only a rough description. In Python, missing this information (e.g. not including the name of each parameter) can cause the function failed to be called in the test (e.g. when arguments are passed by corresponding parameter names).
  3. Data leakage: In Section 3, the authors claimed to crawl the code only from GitHub repositories updated after the training cutoff of GPT-4o, October 2023. But this only limits the repository update time, but not the update time of each function. For example, the `make_grid` function in the dataset was last updated at May 12, 2022, before the cutoff date of GPT-4o, causing a potential data leakage. Even worse, some source repositories of the benchmark are not updated since long ago. For example, `IntelLabs/nlp-architect` was archived on Nov 8, 2022, and the code of `itdxer/neupy` was lastly updated on Apr 5, 2019, `DeepRegNet/DeepReg` was updated on Jun 19, 2021. Overall, the dataset might contain serious data leakage.
  4. Context: This benchmark examines the quality of the generated functions by replacing the original functions. However, the necessary context of the target function is not passed to the LLM.
  5. Executive environment: This benchmark contains calls to external libraries. Different versions of these libraries have different APIs. However, the list of available libraries and their versions are never told to the LLMs. If the LLMs import a library not in the environment, the generated code accidentally fails.

Minor questions/limitations:
1. Section 5.2 Finding 6: The comparison between bug distributions of general code generation and Deep Learning code generation is invalid: the authors compare the failures of different LLMs on different benchmarks. There are two variables to explain the difference in bug distribution, and we do not know how much of the difference comes from the difference in model capabilities, and other portions come from the difference in the coding task.
2. Errors in dataset: the columns ` test_suites` and ` ground_truth_function_code` do not match other columns in the dataset. For example, for the test sample `to_image` in Figure 9, these columns are probably for the function ` distort_points_kannala_brandt`.
3. The file `src/LLM/call.py` in the repository is not runnable, because of the text `W` on line 30.
4. Environment: although there are many dockerfiles, the testing frameworks in the benchmark rely on local environments, and there is no mention of these dockerfiles in project otherwhere. This means there is no environment separation, and the library version conflicts are very easy to happen.
5. The repository is full of unprofessional code, and whether the benchmark actually works on another computer is very questionable.

**Questions:**

See above

---

> ### Author Response · Authors · 2025-11-21
>
> ### Weakness 1: Prompt Quality
> Thank you for pointing this out. We follow a strict multi-step process—conducted by three authors—to manually evaluate and refine prompts (the full procedure is presented in Appendix D). This ensures that each prompt contains complete information, including input/output formats, boundary conditions, and required error handling.
>
> The bug taxonomy is based on general categories, and in this case, some prompts lacked explicit information about which DL framework to use. This is broader contextual information rather than a flaw in the prompt construction. We acknowledge that prompts can be improved by including DL framework details, and we will update the dataset and taxonomy accordingly. Importantly, this category represents only 2.3% (10 instances) of all GPT-4o failures and does not meaningfully impact our findings.
>
> ### Weakness 2: Function Signature
> When executing prompts, we automatically insert the required function signature during preprocessing. This aligns with standard practices in many code-generation benchmarks. Therefore, missing signatures are not the cause of the reported failures. The exact function signature provided to the LLMs is available in the `ground_truth_function` column.
>
> We separate this field so researchers can modify or obfuscate function signatures if desired. In the revision, we will clarify the role of preprocessing and make it explicit that LLMs are provided with all necessary information to generate compilable and runnable code. We will also release the preprocessing script to allow researchers to reuse or adapt the signature-insertion logic.
>
> ### Weakness 3: Data Leakage
> Thank you for raising this concern. We plan to provide a “live” version of the dataset with a configurable cut-off date for each function, which will help mitigate leakage issues. Our current live-version script computes cut-off dates using file modification timestamps. The experiment in Table 2 shows that newer data is generally more challenging—potentially less likely to be leaked.
>
> We acknowledge a small coding mistake that inadvertently included 17 instances modified before the October 2023 cut-off. We will correct this in the final live version. Nevertheless, LLM performance remains low overall, so these 17 instances do not materially affect our findings.
>
> ### Weakness 4: Context
> Thank you for raising this point. To ensure comparability with DS-1000 and alignment with prior work, we intentionally use zero-shot prompting without including entire repository contexts. Each prompt provides the necessary information to generate function-level code.
>
> We also include the precise location of the ground-truth function, allowing future researchers to incorporate additional contextual code if they wish. We will make this explicit in the revision.
>
> ### Weakness 5: Exclusive Environment
> Thank you for pointing this out. Our implementation ensures isolation and reproducibility throughout. Each repository is mapped to a dedicated Dockerfile, and each generation is executed within this isolated environment. Rebuilding the Dockerfile recreates the exact environment we used for evaluation.
>
> Additionally, we apply a post-processing step that imports missing dependencies before execution, preventing failures due to missing or mismatched libraries. We will clarify these details in the revised paper.
>
> ---
>
> ## Response to Questions
>
> ### Q1: Distribution of General Code vs. DL Code
> Thank you for this valid point. As noted, the comparison was meant only to contextualize the error distribution of DL-generated code and is reported only for GPT-4o. It is not part of our core findings. Based on this feedback, we will move this comparison to the appendix to make room for more essential details.
>
> ### Q2: Source Code Error
> Thank you for pointing this out. The issue was a typo introduced when preparing the separate public repository. It was identified and fixed before the rebuttal. We perform internal code reviews and involve external collaborators to verify the repository before official release.
>
> ### Q3: Environment
> Thank you for raising this point. As noted earlier, each repository has its own Dockerfile. During evaluation, we instantiate a fresh Docker environment for each generation, ensuring correct library versions and preventing conflicts. None of the failures we report stem from missing libraries or incorrect versions. This will be made explicit in the revision.
>
> ### Q4: Coding Quality
> Thank you for the feedback. Before submission, external collaborators validated the dataset by following our instructions and successfully reproducing results. We will further improve the repository by conducting an additional code review, updating documentation, fixing minor inconsistencies, and ensuring the dataset is fully ready for public release.

---

### Official Review · Reviewer_eoPa · 2025-11-01

**Soundness:** 3
**Presentation:** 3
**Contribution:** 3
**Rating:** 4
**Confidence:** 4

**Summary:**

This paper introduces DL-Bench, a benchmark dataset of 520 function-level deep learning code generation problems sourced from 30 GitHub repositories. The dataset categorizes problems along three dimensions: DL pipeline stages (pre/post-processing, model construction, training, inference, evaluation), ML tasks (like classification, regression, etc), and input data types (image, text, structured array). The authors evaluate seven state-of-the-art LLMs on DL-Bench, finding significantly lower performance (28.5% average pass@1) compared to the existing DS-1000 benchmark (53.3%). The paper includes quantitative analysis showing DL-Bench occupies a semantically distinct space from related benchmarks, a bug taxonomy specific to DL code generation, and demonstrates potential applications for few-shot prompting improvement.

**Strengths:**

-  The benchmark construction is rigorous with multiple validation steps. The filtering from 2,000+ raw data points to 520 high-quality instances through both automated and manual review (involving four co-authors) ensures quality. The evaluation is comprehensive, testing seven diverse LLMs. Enough points were made to show difference between earlier ML/DL benchmarks like DS 1000 and DL Bench.
- The three-dimensional categorization scheme (pipeline stage × ML task × input data type) is novel and provides actionable insights for targeted improvements. The bug taxonomy specifically for DL-generated code extends prior work meaningfully with new categories like "arithmetic and logical errors" and "performance issues."
- The motivation is clear—existing benchmarks lack coverage of complete DL pipelines. The paper systematically demonstrates four dimensions of divergence from existing benchmarks (semantic space, DL-relevance, solution complexity, and LLM performance).
- The work addresses a real gap. The finding that state-of-the-art models achieve only 28.5% average pass@1 on DL-Bench versus 53.3% on DS-1000 is striking and highlights the challenge and introduces a solid benchmark in the field of DL based code generation problems.

**Weaknesses:**

- There is inconsistency in reporting evaluations with other prior work. For showing that DL bench has no overlap with prior benchmarks, it compared itself with DS-1000 and AICoderEval, whereas for all other ablations (especially to show difficulty and categories), it didn't include AICoderEval.

- With only 520 instances from 30 repositories, the dataset is relatively small, compared to its former dataset DS-1000, it is almost half its size.

- The manual labeling process is critical but under-documented, the claim of "strong inter-rater reliability" in Section 7 lacks supporting evidence. The same can be said about labels in the classification task.

- There is no comparison with human evaluations to know how well do humans do on this task. That will genuinely tell more about the diversity and difficulty of the dataset.

**Questions:**

- Figure 1 can be more clearer with proper annotations showing where a LLM is involved and where humans are involved.
- There has been very less discussion about test cases / test case quality.
- Why does O3-Mini show lower performance with three-shot prompting on DS-1000 (61.0% to 50.2%)? This seems counterintuitive.
- What specific criteria led to the reduction from 2,000 raw data points to 520 final instances? It was not clear while reading the paper.
- The abstract can be made shorter and clearer, especially while discussing the effectiveness of the benchmark.

---

> ### Author Response · Authors · 2025-11-21
>
> ### Weakness 1: Inconsistency in Including AICoderEval Across Different Comparisons
> We thank the reviewer for pointing this out. Our goal was to compare DL-Bench with all relevant ML/DL-focused benchmarks; however, AICoderEval could not be included in several analyses because its full dataset and evaluation scripts were not publicly available. This prevented us from computing consistent pass@k scores, bug taxonomy labels, and per-category breakdowns. In the revised version, we will explicitly clarify this limitation and explain why DS-1000 is the only benchmark used across all ablations.
>
> ### Weakness 2: Dataset Size Relative to DS-1000 (520 Instances)
> We appreciate the reviewer’s observation. DL-Bench was designed with strict filtering criteria to ensure that each instance is complete, testable, and genuinely DL-relevant. Many of the initial 2,000+ samples were removed due to missing or incomplete tests, trivial or boilerplate code, ambiguous behavior, or lack of DL relevance. The resulting size of 520 instances is comparable to—and even larger than—several popular datasets, such as HumanEval (164 instances), MLE-Bench (72 instances), and AICoderEval (492 instances). We also plan to expand DL-Bench through additional “live” versions while maintaining strict quality standards.
>
> ### Weakness 3: Manual Labeling Process Is Under-Documented
> We thank the reviewer for raising this concern. Due to space limitations, Appendix C.2 reports the inter-rater agreement score of 0.83, indicating strong agreement. In the revision, we will highlight this score more prominently in the main text under Section 3.
>
> ### Weakness 4: Lack of a Human Evaluation Baseline
> We appreciate the reviewer’s suggestion. Human-expert evaluation is indeed valuable for contextualizing dataset difficulty. Due to the complexity and subjectivity of human evaluations, we follow prior work by comparing DL-Bench with existing benchmarks through their relative performance when evaluated by LLMs. In the revision, we will explicitly acknowledge this as a limitation.
>
> ---
>
> ## Response to Questions
>
> ### Q1: Figure 1 Clarity (LLM Involvement vs. Human Involvement)
> We appreciate the feedback. In Figure 1, the LLM was involved only in generating the initial prompt at step 7. In the revision, we will update the figure to clearly label this step as “LLM” rather than using the GPT logo.
>
> ### Q2: Limited Discussion of Test Cases and Test Case Quality
> All test cases originate from the source GitHub repositories and reflect real usage patterns. We designed the filtering steps to include only top-starred repositories with high-quality, practical test cases. In the revision, we will emphasize this point more clearly.
>
> ### Q3: Why Does O3-Mini Perform Worse With Three-Shot Prompting on DS-1000?
> This outcome may indicate that providing examples does not consistently improve O3-Mini’s performance. Notably, after incorporating stage prediction guidance, the performance improved (from 50.2% to 54.3%). This demonstrates the usefulness of the categorization provided by DL-Bench—even when applied to a different dataset such as DS-1000. In the revision, we will clarify this observation and highlight that these few-shot experiments are exploratory in nature, intended to showcase the potential of DL-Bench’s structure.
>
> ### Q4: Criteria for Reducing 2,000 Raw Samples to 520 Final Instances
> We thank the reviewer for the question. Due to space limitations, much of this detail appears in the appendices—for example, Appendix C.1 describes the filtering steps. In the revision, we will move high-level descriptions of these steps into the main text or more explicitly refer readers to the appendix.
>
> ### Q5: Abstract Clarity
> We appreciate this suggestion. We will revise the abstract to make it more concise and to clearly emphasize the key contributions and empirical findings.

---

### Official Review · Reviewer_eF5M · 2025-11-01

**Soundness:** 3
**Presentation:** 3
**Contribution:** 3
**Rating:** 6
**Confidence:** 4

**Summary:**

This paper introduces DL-Bench, a benchmark designed to address the inadequacies of existing benchmarks in evaluating the DL code generation capabilities of LLMs. Current benchmarks focus primarily on fragmented data science code snippets and fail to comprehensively cover the entire DL workflow. DL-Bench fills this gap by providing a function-level code generation benchmark that spans various DL stages, multiple task types, and different data types. The core contributions are: 1) The construction and release of a more challenging, DL-focused code generation benchmark; 2) A fine-grained analysis of LLM performance across different DL sub-tasks, enabled by a multi-dimensional taxonomy; 3) The proposal of a bug taxonomy specifically for LLM-generated DL code; 4) A demonstration of how the benchmark can be used to improve code generation performance on other datasets.

**Strengths:**

1. Clear and Significant Problem Statement. The paper identifies the limitations of existing data-science-oriented benchmarks in assessing DL code generation, addressing a clear gap in the current field.
2. Well-Reasoned Benchmark Design. The three-dimensional taxonomy (DL pipeline stages, ML task types, input data types) is a commendable design choice. It enables a more in-depth and fine-grained analysis of model capabilities than previously possible.
3. Comprehensive Experimental Evaluation. The benchmark is validated across seven SOTA LLMs under controlled conditions, with statistical rigor. The results demonstrate that DL-Bench is semantically distinct and requires more complex solutions than prior benchmarks.

**Weaknesses:**

1. Support for Some Conclusions is Weak.
	1) "Live" Version Evaluation: The results in Table 2 are inconclusive, as the cutoff date becomes more recent, the performance degradation for some LLMs is not significant, and Qwen Coder 2.5's performance even increases. This weakens the conclusion that newer data is inherently more challenging.
	2) Classifier Performance Discrepancy: The classifier used to predict pipeline stages reportedly achieves high accuracy on manually labeled instances but a low weighted F1-score of only 0.56 in five-fold cross-validation. This discrepancy is not adequately explained and impacts the reliability of the Stage-Predicted Three-Shot experiments.
2. The paper suffers from several writing and formatting issues. For instance, there are missing spaces after commas, and punctuation is sometimes absent where required (e.g., a missing period before "Finally, we" in Section 4.1).

**Questions:**

See Weaknesses

---

> ### Author Response · Authors · 2025-11-21
>
> ### Weakness 1: Support for Conclusions Regarding the “Live” Versions
> We thank the reviewer for this insightful comment. We agree that Table 2 shows mixed trends across models and that the performance drop is not uniform. This variability may stem from multiple factors unrelated to the cut-off date or data leakage, and investigating these factors lies outside the scope of our work. Our primary intention was to show that we can—and will—provide these live versions, offering an additional perspective for researchers interested in studying the effect of data leakage in future work. In the revision, we will clarify the phrasing to avoid overstating this result and will explicitly note that the evidence suggests a trend but is not conclusive. We will also add per-model effect sizes to strengthen the transparency of this analysis.
>
> ### Weakness 2: Classifier Performance Discrepancy (Manual Accuracy vs. Weighted F1)
> We appreciate the reviewer highlighting this point. The classification accuracy of the random sample from DS-1000 is high because most (86%) of the 100 sampled instances belong to the pre/post-processing stage, which is easier to predict. In contrast, cross-validation was performed on DL-Bench, which contains a more diverse label distribution (shown in Table 4), including categories that are harder to predict. Below is the per-class precision for the five labels in the DS-1000 sample:
>
> | Dataset | Pre/Post Processing | Model Construction | Training | Inference | Evaluation |
> |---------|----------------------|--------------------|----------|-----------|------------|
> | DS-1000 (100 samples) | 0.988 | 0.857 | 0.667 | 0.500 | NA |
>
> As shown, pre/post-processing achieves far higher precision, whereas Training and Inference remain much more challenging. In the revision, we will explicitly explain this discrepancy, report per-class precision to give a clearer view of performance across categories, and discuss how this affects the Stage-Predicted Three-Shot experiment. Importantly, this few-shot experiment is exploratory and meant to demonstrate a potential application of the dataset.
>
> ### Weakness 3: Writing and Formatting Issues
> We thank the reviewer for noting these writing inconsistencies. We will conduct a thorough proofreading pass to correct stylistic and punctuation issues, including missing spaces, inconsistent comma usage, and missing periods (such as the one before “Finally, we” in Section 4.1). We will ensure that the final version follows a consistent formatting and writing standard.

---

### Official Review · Reviewer_F6LC · 2025-11-01

**Soundness:** 3
**Presentation:** 2
**Contribution:** 3
**Rating:** 8
**Confidence:** 4

**Summary:**

This manuscript presents DL-Bench, a benchmark to evaluate LLMs on DL function code generation. The benchmark is sourced from GitHub repositories, includes generated prompts for extracted DL functions, and is categorized based on sample pipeline step, task type, and input type. The authors show that DL-Bench differentiates itself from related datasets through a semantic comparison, token analysis, and code complexity analysis. When evaluating state-of-the-art LLMs on DL-Bench, the overall performance is worse than related datasets and unique types of failures were identified. Furthermore, the authors show that DL-Bench can be used for purposes other than benchmarking, such as DL code classification and few-shot prompting.

**Strengths:**

Overall, the manuscript is very well structured and written. The findings are substantiated with significant results from relevant experiments and presented nicely in figures and tables.

**Weaknesses:**

In my opinion, there are a few minor weaknesses in the manuscript. The first is that the text is incredibly dense and still many details had to be left out of the main text and added as an appendix. For example, the data categories and labels are only listed in Table 3 and in Appendix F but not in the main text. This has the effect that I sometimes didn't feel I grasped all the important aspects of the benchmark as I was reading the text. I am not sure how this can be best remedied within the page limit, but I did feel sometimes that the gray "Findings" text bubbles were somewhat repetitive in regards to the text that come right before them. Additionally, I spotted a few small language errors in the text. These include inconsistency between how figures are referenced in the text (sometimes "Figure X" is used while "Fig X" is used elsewhere), repeated author names in text references (e.g., "Shin et al.Shin et al" on line 98), and other miscellaneous errors (e.g., "LLM" -> "LLMs" on line 179, "a commonly used *metric* pass@k" on lines 244-245, "Also, even if" -> "Also, even though" on line 466).

**Questions:**

These weaknesses do not significantly detract from the manuscript's merit.
Therefore, I would recommend accepting this manuscript. I hope the authors can make minor adjustments to their manuscript to address the weaknesses I explained above.

---

> ### Author Response · Authors · 2025-11-21
>
> ## Limitations and Weaknesses
>
> ### 1. Text Density and Design Details
> We thank the reviewer for this helpful observation regarding the presentation of the dataset construction. Since we aim to disclose as many details as possible to promote transparency and reproducibility, we chose to include lower-level details in the appendix. In the revision, we will ensure that the main text provides clearer high-level descriptions so readers can understand the process before consulting the appendix for finer details when needed.
>
> ### 2. Repetition Between Narrative Text and “Findings” Boxes
> We appreciate the reviewer pointing this out. The purpose of the “Findings” boxes is to summarize key results, but we agree that some overlap currently exists. We will revise these boxes to focus solely on concise takeaways and remove repetitive phrasing so that they complement—rather than duplicate—the surrounding text.
>
> ### 3. Minor Language Inconsistencies and Typographical Issues
> We thank the reviewer for highlighting these issues. We will perform a thorough proofreading pass to correct inconsistencies in figure referencing (e.g., using either “Figure” or “Fig.” consistently), remove duplicated author names (e.g., “Shin et al. Shin et al.”), correct pluralization (e.g., “LLM” → “LLMs”), and revise the noted sentences for clarity. We will ensure the writing adheres to a consistent technical style.
>
> ## Response to Questions
> We appreciate the reviewer’s positive evaluation and recommendation for acceptance. We will incorporate all suggested adjustments—clarifying category descriptions in the main text, reducing repetition in the findings boxes, and correcting minor language issues. These revisions will improve the readability and presentation of the manuscript without altering its core contributions.

---

### Meta-Review · Area_Chair_9Vf6 · 2026-01-13

**Summary:**

This submission introduces DL-Bench, a benchmark for deep-learning–specific function-level code generation, accompanied by extensive analyses and evaluations across multiple LLMs. Several reviewers find the problem well-motivated and acknowledge solid engineering effort, clear taxonomy design, and useful empirical analyses. Two reviewers recommend acceptance, while others are mixed to negative, citing concerns about benchmark quality, experimental validity, and presentation. The strongest criticism questions the benchmark’s quality and validity, not just its difficulty or novelty. While DL-Bench is shown to be harder and semantically distinct from prior benchmarks, reviewers argue this does not automatically imply higher quality. such as prompt completeness and unclear or inconsistently exposed function signatures. Further considering a potential violation of the double-blind review policy in the provided Kaggle Dataset, ACs recommend reject.

**Reviewer Concerns:**

The strongest criticism questions the benchmark’s quality and validity, not just its difficulty or novelty. While DL-Bench is shown to be harder and semantically distinct from prior benchmarks, reviewers argue this does not automatically imply higher quality. such as prompt completeness and unclear or inconsistently exposed function signatures. And potential violation of the double-blind review policy.

**Reviewer Scores:**

Two reviewers recommend acceptance, while others are mixed to negative, citing concerns about benchmark quality, experimental validity, and presentation. The ratings are 2, 4, 6, 8.

---

### Decision · Program_Chairs · 2026-01-26

Reject